# SWE-rebench: An Automated Pipeline for Task Collection and Decontaminated Evaluation of Software Engineering Agents

**Ibragim Badertdinov**\*
Nebius

**Alexander Golubev**\*
Nebius

**Maksim Nekrashevich**
Nebius

**Anton Shevtsov**
Nebius

**Simon Karasik**
Nebius

**Andrei Andriushchenko**
Nebius

**Maria Trofimova**
Nebius

**Daria Litvintseva**
Nebius

**Boris Yangel**
Nebius

## Abstract

LLM-based agents have shown promising capabilities in a growing range of software engineering (SWE) tasks. However, advancing this field faces two critical challenges. First, high-quality training data is scarce, especially data that reflects real-world SWE scenarios, where agents must interact with development environments, execute code and adapt behavior based on the outcomes of their actions. Existing datasets are either limited to one-shot code generation or comprise small, manually curated collections of interactive tasks, lacking both scale and diversity. Second, the lack of fresh interactive SWE tasks affects evaluation of rapidly improving models, as static benchmarks quickly become outdated due to contamination issues. To address these limitations, we introduce a novel, automated, and scalable pipeline to continuously extract real-world interactive SWE tasks from diverse GitHub repositories. Using this pipeline, we construct SWE-rebench, a public dataset comprising over 21,000 interactive Python-based SWE tasks, suitable for reinforcement learning of SWE agents at scale. Additionally, we use continuous supply of fresh tasks collected using SWE-rebench methodology to build a contamination-free benchmark for agentic software engineering. We compare results of various LLMs on this benchmark to results on SWE-bench Verified and show that performance of some language models might be inflated due to contamination issues.

## 1 Introduction

Large Language Models (LLMs) have demonstrated impressive capabilities in SWE tasks, including code generation, debugging, and automated development workflows. Building on these capabilities, researchers have begun creating LLM-driven agents that interact with real codebases and development environments, performing actions and receiving feedback [Jin et al., 2025]. While frontier proprietary models drive the performance of the most competitive agents (e.g., OpenHands [Wang et al., 2024], Moatless Tools [Antoniades et al., 2024], Agentless [Xia et al., 2024]) on key benchmarks like SWE-bench [Jimenez et al., 2024], there exists a significant opportunity to enhance open-source models [Wang, 2025, Yang et al., 2025a, Wang et al., 2025, Aggarwal et al., 2025, Ma et al., 2025,

---

\*Equal contribution. Correspondence to `ibragim-bad@nebius.com`

39th Conference on Neural Information Processing Systems (NeurIPS 2025) Track on Datasets and Benchmarks.

Wei et al., 2025, Golubev et al., 2024]. Progress in this direction, particularly toward complex agentic behaviors, may be accelerated with access to large-scale, high-quality training data that mirrors the interactivity inherent in real-world software development. Existing powerful open-source models like DeepSeek-V3 [DeepSeek-AI, 2024], LLaMa 4 [Meta AI, 2025] and Qwen3 [Team, 2025] could potentially be fine-tuned to achieve comparable performance in specific SWE domains, but this hinges on the availability of suitable interactive task data.

Current approaches to training LLMs for programming often rely on code data from open-source repositories [Lozhkov et al., 2024] or synthetic instruction datasets [Wei et al., 2024] that are used for instruction tuning. However, training robust software engineering agents for real-world scenarios necessitates datasets that extend beyond simple code generation. To truly enable learning through methods like Reinforcement Learning (RL), which thrives on trial-and-error, agents require interactive tasks coupled with automatic verification mechanisms. Such data must allow agents to perform diverse actions, observe environment responses after each step, and receive eventual verification outcomes that determine task success. Unlike domains such as mathematics [Shao et al., 2024] or web navigation [Pan et al., 2024a], software engineering has historically lacked such large-scale interactive datasets due to the complexities of configuring diverse, executable environments at scale. While recent efforts like SWE-Gym [Pan et al., 2024b] and SWE-PolyBench [Rashid et al., 2025] represent promising steps, their manual curation processes and reliance on a limited number of repositories constrain their scope, diversity, and scalability.

Furthermore, the evaluation of rapidly advancing LLM-based agents also faces significant challenges. Static benchmarks, while initially valuable, can become compromised by data contamination as newer models become exposed to test instances during their extensive pre/post-training. Moreover, the lack of standardized evaluation protocols, variability in agent scaffolds and inconsistent reporting practices make direct comparisons between models difficult and can obscure their true capabilities.

To address these challenges in both training data availability and evaluation reliability, we present a scalable, fully automated pipeline for continuous collection of software engineering tasks from real-world GitHub repositories. Building upon our prior work such as SWE-bench Extra [Badertdinov et al., 2024], which has been well-received by the community and is already used to train open-source software engineering agents [Wang et al., 2025], our approach eliminates manual intervention and significantly expands task diversity and scale. To the best of our knowledge, this is the first system enabling fully automated, scalable collection of executable tasks from a wide set of real-world repositories, specifically designed to support interactive agent training and robust benchmarking.

Our main contributions are as follows:

- A scalable and fully automated pipeline for mining real-world software engineering tasks from GitHub, covering environment configuration, build setup, and test validation.
- SWE-rebench[2], a public dataset of more than 21,000 interactive Python-based SWE tasks, designed to train and benchmark agents in diverse executable environments, particularly suitable for reinforcement learning-based approaches.
- A public SWE-rebench leaderboard[3] that offers continuously updated, decontaminated, and standardized evaluations for LLM-based agents, promoting transparency and fair comparisons across both open- and closed-source models. As the leaderboard is continuously updated, the current set of evaluated models can be found on the leaderboard website.

By focusing on scale and automation, SWE-rebench aims to fill a critical gap in the LLM agent ecosystem. We believe it will serve as a foundational resource for accelerating open-source research and improving the reliability and performance of LLM-based software engineering agents.

## 2   An automated pipeline for collecting software engineering tasks

In this section we describe our automated pipeline for mining verifiable software engineering tasks at scale, that we used to build SWE-rebench, a dataset of 21,336 verifiable SWE tasks from 3468 distinct GitHub repositiories. Our pipeline comprises four stages: *preliminary task collection*,

---

[2]Available on Hugging Face Datasets: https://huggingface.co/datasets/nebius/SWE-rebench
[3]Accessible at: https://swe-rebench.com/leaderboard

*automated installation instruction configuration*, *execution-based installation verification*, and *quality assessment*, which are fully described in this section. While our methodology incorporates several techniques from SWE-bench, it also introduces innovations to enhance automation and scalability. We detail the distinctions and novel aspects of our approach compared to the original SWE-bench methodology in Appendix G. The computationally intensive nature of our pipeline is managed through a distributed storage and computing platform TractoAI [TractoAI, 2025], which provides capabilities for efficient parallel processing and data management, helping us optimize throughput of each stage to enable rapid reprocessing whenever we change the pipeline.

## 2.1 Preliminary task collection

In the first stage, we download raw input data from multiple origins, merge them, and perform preliminary filtering. The primary sources for our data are GitHub Archive [Grigorik, 2011] and GitHub.

- **GitHub Archive.** The GitHub Archive is a major source of public events on GitHub. Each day, it publishes a JSON archive listing all GitHub events from that day. We use this archive to collect detailed data about issues: issue description, discussion, linked pull requests, and metadata such as creation date and labels. We also extract information about pull requests, including their merge status, last commit, and discussions.

- **GitHub.** We clone relevant GitHub repositories with their full commit histories to our local storage. A local copy enables efficient access to repository data and helps avoid GitHub API rate limits. We use preserved commit history to identify changes associated with pull requests and perform version analysis for automated dependency setup in later stages.

To initiate the dataset building process, we download approximately 450,000 pull requests linked to issues created before May 1, 2025. These originate from over 30,000 repositories that feature permissive licenses granting broad usage rights (see Appendix D for the list of included license types) and where Python constitutes over 75% of the codebase lines of code.

We then link issues with pull requests that mention resolving them in their title or description, applying filters to select instances where:

- The issue is from a Python repository with a permissive license.
- The issue is marked as resolved.
- The PR is merged into the main branch.
- The PR is not linked to multiple issues.
- The issue description is longer than 10 characters.
- The PR must introduces or modifies tests and includes code changes beyond test files.
- Changes affect 1 to 15 files.

This filtering aims to eliminate unsuitable candidates, particularly those lacking tests. We require pull requests that introduce or modify tests, as these are crucial for automatically evaluating whether a proposed code patch resolves the described issue. For each selected pull request, the overall patch is divided into two components: a solution patch, containing changes to non-test files intended to address the issue, and a test patch, comprising only changes to test files. After applying all filtering criteria, approximately 153,400 potential task instances remain.

## 2.2 Automated installation instructions configuration

Datasets like SWE-bench [Jimenez et al., 2024] or SWE-Gym [Pan et al., 2024b] rely on manual curation to configure executable environments for each repository. This approach inherently limits scalability, often confining such datasets to a small selection of well-known repositories. Key steps typically include project versioning (mapping multiple task instances to a single valid environment) and defining setup instructions (to install dependencies and run tests). Manually conducting these steps on a large-scale, diverse task collection is infeasible; therefore, we employ a fully automated approach.

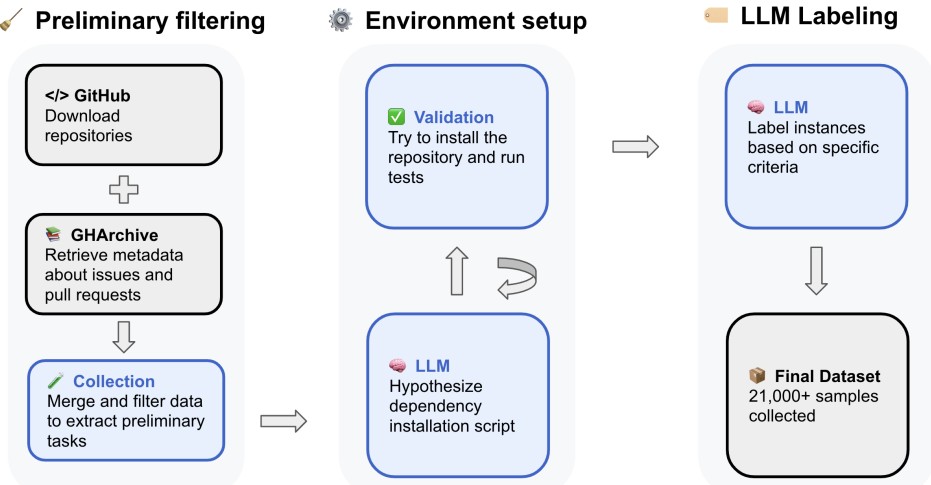

Figure 1: Overview of the automated pipeline for collecting software engineering data.

After preliminary filtering described in Section 2.1, remaining issues are treated as task instances. We group these task instances by project versions inferred from `git tag` outputs, normalizing versions to `major.minor` format (e.g., `1.2.3` is normalized to `1.2`). For each version group, we select the `base_commit` of the pull request linked to the task instance with the most recent `base_commit` date. We prioritize this most recent `base_commit` because developers typically maintain dependency compatibility within minor versions, and later commits often include important environment fixes. This approach generally provides a stable dependency set, often sufficient for executing test patches from all tasks in that group within a shared environment. The `git tag` command provides a version for approximately 95% of task instances. We assign a unique version to the rest of the tasks, so that each one of them uses its own environment. We employ an agentless approach, inspired by [Xia et al., 2024], to generate candidate environment setup instructions. This process involves several LLM-driven steps:

- **Identifying relevant files:** An LLM scans repository files (e.g., `README.md`, `Dockerfile`, `setup.py`) to find potential sources of installation information.

- **Extracting installation recipe:** The LLM processes the concatenated content of files identified in the previous stage to produce a structured JSON object detailing the installation recipe. Files are provided to the LLM in the format: `<filename>F.ext</filename>\n<content>CONTENT</content>`. An example of the LLM's reasoning and the resulting JSON recipe is provided in Appendix B.3.

We use the `Qwen2.5-72B-Instruct` model [Qwen et al., 2025] (prompt in Appendix B.2) to generate up to three candidate JSON recipes per task. If an error occurs during the subsequent scripted installation or test execution (derived from a recipe), the LLM attempts to refine that recipe by analyzing error logs and the original instructions (see correction prompt in Appendix B.4). This iterative refinement enables successful environment configuration for tasks with issues like missing libraries or incorrect setups, allowing their inclusion in the final dataset. Our approach successfully produces a working installation recipe for at least one task in 31% of all repositories. We also catalog the primary installation failure reasons (e.g., unpinned dependencies or flaky tests) and provide a brief review in Appendix B.5.

We also explored dependency installation using an interactive agent that directly interacts with a Docker environment to install projects and run tests. While this interactive agent occasionally configured environments more effectively, it proved to be significantly more resource-demanding. The chosen agentless method is more computationally efficient for large-scale processing, and generating multiple candidate recipes can further improve its effectiveness, making it our primary approach. A comparative evaluation of these approaches on a curated subset of SWE-bench tasks is detailed in Appendix C.

## 2.3 Execution-based installation verification

To confirm task solvability and the integrity of the provided tests, we perform execution-based installation verification. This stage involves installing the environment for each task within a container and executing the tests from the pull request's test patch. We then parse the test run outputs to ensure that: (1) at least one test from the test patch fails before applying the solution patch (i.e., changes to non-test files from the original pull request), (2) all tests from the test patch that initially failed subsequently pass after the solution patch is applied, and (3) any tests from the test patch that initially passed continue to pass after the solution patch is applied. Tasks are considered valid only if they meet these conditions.

Processing numerous task instances, each potentially with multiple candidate recipes requiring installation, testing, and logging, necessitates distributed execution to manage the workload efficiently. We use TractoAI for this purpose, as it enables distributed container building and parallel execution of verification tasks in built containers across a cluster.

For installation verification, we use a default base container with pre-installed basic dependencies (e.g., **conda**, **gcc**). Our own image registry and internal PyPI/APT mirrors help cache popular dependencies, accelerating container launches and reducing reliance on external sources. During task verification we perform the following steps:

- Install project dependencies in an isolated container using **buildah**. We utilize **tmpfs** for file system operations to minimize disk I/O and accelerate builds.
- Execute tests and parse logs to identify tests validating the solution.
- Build final container images upon successful verification.
- Record exact dependency versions (via `pip freeze`, `conda env export`) after a successful setup to mitigate reproducibility issues from unpinned dependencies in Python projects, ensuring consistent environment recreation in the future.

## 2.4 Automated instance quality assessment

For our collected tasks to be effectively used for reinforcement learning, they should possess certain properties; otherwise, RL agents might generate trajectories that appear as failures but are actually due to task imperfections (e.g., an underspecified issue making the task unsolvable, or flawed tests that a correct solution cannot pass), leading to incorrectly penalizing the agent. While SWE-bench Verified ensures these properties through manual verification, the scale of our collection necessitates an automated approximation of these checks. To assess these properties automatically, we fine-tune an instruction-following model using human annotations from SWE-bench Verified to predict:

- **Issue Clarity:** Whether the GitHub issue description is sufficiently detailed for a developer to understand and solve the problem.
- **Task Complexity:** The estimated effort to resolve the issue, considering reasoning, code modification, and codebase familiarity.
- **Test Patch Correctness:** Whether tests in the pull request accurately verify the intended fix without over-reliance on specific implementation details.

We fine-tune Qwen 2.5-72B-Instruct using annotations from SWE-bench Verified. For each of the over 3,800 examples, the model receives the issue description, the canonical solution patch, and the test patch as input. It is then prompted to predict one of three binary quality labels: **Issue Clarity**, **Task Complexity**, or **Test Patch Correctness**. We train the model to predict each label independently; each task instance is assessed for each quality characteristic separately using a 75/25 training/validation split (total 413 validation examples).

For **Task Complexity** (where 'high-score' implies >1 hour to solve; 100 high-score vs. 313 low-score examples in validation), our fine-tuned model achieved 81% accuracy and a weighted F1-score of 0.82. This is an improvement over the baseline Qwen-72B-Instruct, which achieved 68% accuracy. For **Test Patch Correctness** (180 high-score vs. 233 low-score examples), the model achieved 67% accuracy (weighted F1: 0.65). For **Issue Clarity** (84 high-score vs. 329 low-score examples), it achieved 79% accuracy (weighted F1: 0.76). A more detailed prediction quality analysis, including precision and recall per class, can be found in Appendix F, Table 4.

The LLM-generated labels for **Issue Clarity**, **Task Complexity**, and **Test Patch Correctness** are provided as metadata with each task instance. While this automated assessment is not perfect, these labels offer users a means to filter the dataset and select task instances according to their specific criteria. For example, these labels facilitate task difficulty control more precise than heuristics like the number of modified files (used, for example, for SWE-bench Lite subset), as the number of changed files can be misleading: a multi-file change might be simple (e.g., a repeated parameter update), while a single-file change might lack clear issue descriptions or adequate tests for full validation. Thus, these LLM-based scores for difficulty and clarity of description and tests empower users to perform more nuanced task selection, helping them identify challenging yet solvable and clearly specified tasks beneficial for their specific model training or evaluation needs, and potentially aiding in mitigating benchmark saturation.

This four-stage pipeline automates the collection and processing of interactive software engineering tasks. The process yields the SWE-rebench dataset of 21,336 annotated task instances, which is publicly available on Hugging Face Datasets. Accompanying code for utilizing the dataset, including scripts for tasks evaluation, is available on GitHub. An example of a task instance with its full annotation is provided in Appendix E.

## 3 SWE-rebench benchmark

In this section we discuss key limitations of existing evaluation setups for LLM-based software engineering agents and how our automated data pipeline described in Section 2 helps to address them. We leverage this pipeline to construct SWE-rebench, a benchmark built from hundreds of real-world, executable SWE tasks. It comprises 294 executable tasks from 169 diverse repositories, selected using filtering criteria detailed in Appendix H, and is part of the broader SWE-rebench dataset release. To ensure reliable and standardized evaluation, we maintain a private leaderboard based on this benchmark.

### 3.1 Challenges in SWE agent benchmarking

We identified the following key areas for improvement:

- **Potential data contamination:** SWE-bench, the de facto evaluation standard for SWE agents, has been public since late 2023. Models released afterward may have been exposed to its data during training, risking inflated scores and confounding generalization with memorization.

- **Incomparable results due to scaffolding variability:** Current evaluation practices allow for a wide range of setups. Performance on SWE-bench is often heavily influenced by highly engineered prompts, complex multi-agent frameworks, retry mechanisms, best-of-N sampling strategies and validation loops. While these techniques demonstrate the potential of systems built around LLMs, they make it difficult to isolate and compare raw capabilities of different LLMs. Furthermore, the scaffoldings are often developed and tuned on subsets from SWE-bench, inadvertently leading to a potential for implicit overfitting to the benchmark's specific characteristics.

- **Lack of standardized and verifiable evaluation:** SWE-bench results are typically performed and reported by individual teams. This decentralized approach lacks a mechanism for independent verification and can potentially lead to inconsistencies or misleading reporting practices such as reporting pass@N as pass@1 or implicitly using information derived from final tests. The reliance on closed-source frameworks for many submissions further reduces the transparency and reproducibility of the evaluation process.

- **High variance in agent performance across runs:** Due to the stochastic nature of agent trajectories, the outcome of a single run can vary significantly. This includes cases where a model may successfully generate correct actions or recover from mistakes in some runs, but fail to do so in others. Without averaging or reporting performance across multiple runs, the results can be unrepresentative. In particular, evaluating an agent multiple times and reporting only the best-performing run risks overstating the model's actual capabilities and resolved rate.

## 3.2 Principles of SWE-rebench

SWE-rebench is designed to address the above challenges and support rigorous, model-centric evaluation through several core principles:

- **Centralized and standardized evaluation framework:** All evaluations on SWE-rebench are conducted by our team by using a fixed scaffolding, i.e., every model is assessed by using the same minimal ReAct-style agentic framework [Yao et al., 2023], identical prompts and default generation hyperparameters as recommended by model developers. We standardize the context length to 128K tokens for all evaluations, unless a model only supports a shorter context.

  This strict standardization ensures an equal environment, allowing for direct comparison of the core abilities of different models to understand and solve SWE tasks within a defined, general-purpose interaction structure. While model-specific tuning or a different scaffolding could potentially yield higher scores for a given model, our focus is on establishing a reliable baseline of model capabilities in a common setting.

  It's important to note that the interaction with the development environment is based on the model generating textual commands according to the interaction format described in the prompt. To equalize evaluations, we don't use the function-calling functionality that some of the tested models support. For transparency, we share the exact system prompt used for all model evaluations in Appendix I.

- **Continuous dataset updates and decontamination:** SWE-rebench uses an automated pipeline from Section 2 for a continuous supply of fresh tasks. Since we precisely track the creation dates of the issues and their corresponding pull requests against model release dates, we can explicitly mark potentially contaminated evaluations that include issues created before a model's release date. These evaluations are explicitly marked on our leaderboard, to ensure transparency around possible data leakage.

- **Accounting for stochasticity in agent behavior:** To capture performance variability, we run each model five times on the full benchmark. We additionally report both the standard error of the mean (SEM) and pass@5 metrics to provide a statistically grounded and more reliable assessment of each model performance.

This standardized approach allows SWE-rebench to focus on measuring two fundamental aspects of model performance:

- The ability to comprehend a real-world software issue (presented as a GitHub issue), devise a plan, implement a correct code patch, and potentially validate the solution.

- The ability to follow instructions and operate within a structured agentic framework, which is represented by our ReAct scaffolding.

## 3.3 Result analysis

We leverage the decontaminated nature of SWE-rebench to analyze performance trends over time and identify potential signs of contamination effects in prior benchmarks. Specifically, we evaluate models on two distinct temporal subsets of tasks: those created in January 2025 and those from March–April 2025. Table 1 presents model performance across these time windows.

To investigate potential overfitting to the SWE-bench Verified dataset, we compare model performance on SWE-rebench tasks to the same models' performance on SWE-bench Verified. This comparison focuses on open-source models released in 2024 or early 2025, for which the risk of data leakage from the Verified subset is higher. Table 2 summarizes the comparative results on SWE-bench Verified and the March-April 2025 slice of SWE-rebench.

The results from this evaluation showcase several notable observations:

- GPT-4.1 is the only model, which performance noticeably declined on the March–April subset compared to the January subset.

- LLaMa-4-Maverick exhibits a high pass@5 score relative to models with similar mean resolution rates, yet has a relatively modest resolution rate. This indicates that while the

Table 1: Comparison of model performance on SWE-rebench Jan 2025 and SWE-rebench (Mar–Apr 2025). All metrics are reported in percentages. Models released after 1st of March 2025 are denoted with an asterisk (*).

| Model | SWE-rebench (Jan) | | | SWE-rebench (Mar–Apr) | | |
| --- | --- | --- | --- | --- | --- | --- |
| | Resolved | SEM | Pass@5 | Resolved | SEM | Pass@5 |
| gpt-4.1-2025-04-14* | 31.1 | 0.79 | 44.4 | 26.7 | 1.09 | 39.0 |
| DeepSeek-V3-0324* | 21.7 | 1.64 | 35.0 | 21.3 | 0.98 | 32.4 |
| DeepSeek-V3-1226 | 19.1 | 0.58 | 33.3 | 21.9 | 1.44 | 31.4 |
| Qwen3-235B-A22B no-think* | 15.2 | 1.76 | 29.9 | 16.6 | 0.93 | 25.7 |
| Qwen3-235B-A22B think* | 13.7 | 1.81 | 29.9 | 12.2 | 1.33 | 25.7 |
| Qwen3-32B no-think* | 13.2 | 1.17 | 23.9 | 13.7 | 1.03 | 26.7 |
| Qwen3-32B think* | 11.8 | 0.99 | 22.2 | 11.2 | 0.56 | 21.0 |
| Llama-4-Maverick-Instruct* | 8.5 | 0.90 | 20.5 | 12.2 | 1.69 | 27.6 |
| Qwen2.5-72B-Instruct | 8.2 | 0.79 | 18.8 | 9.3 | 1.26 | 19.0 |
| Llama-3.3-70B-Instruct | 8.2 | 0.44 | 15.4 | 11.2 | 0.47 | 22.9 |
| Llama-4-Scout-Instruct* | 5.0 | 0.63 | 12.8 | 5.3 | 0.38 | 14.3 |
| gemma-3-27b-it* | 4.3 | 0.81 | 9.4 | 4.8 | 0.30 | 10.5 |
| Qwen2.5-Coder-32B-Instruct | 2.7 | 0.57 | 7.7 | 3.2 | 0.77 | 9.5 |

Table 2: Comparison of model performance on SWE-bench Verified and SWE-rebench (Mar–Apr 2025). All metrics are reported in percentages.

| Model | SWE-bench Verified | | | SWE-rebench (Mar–Apr) | | |
| --- | --- | --- | --- | --- | --- | --- |
| | Resolved | SEM | Pass@5 | Resolved | SEM | Pass@5 |
| DeepSeek-V3-0324 | 39.7 | 0.35 | 57.4 | 21.3 | 0.98 | 32.4 |
| DeepSeek-V3-1226 | 35.2 | 0.52 | 51.0 | 21.9 | 1.44 | 31.4 |
| LLaMA-3.3-70B-Instruct | 18.1 | 0.66 | 31.6 | 11.2 | 0.47 | 22.9 |
| LLaMA-4-Maverick-Instruct | 16.0 | 0.79 | 39.2 | 12.2 | 1.69 | 27.6 |
| Qwen2.5-72B-Instruct | 11.3 | 0.35 | 27.8 | 9.3 | 1.26 | 19.0 |
| LLaMA-4-Scout-Instruct | 8.8 | 0.48 | 22.2 | 5.3 | 0.38 | 14.3 |
| Qwen2.5-Coder-32B-Instruct | 4.9 | 0.46 | 16.0 | 3.2 | 0.77 | 9.5 |

model can produce correct solutions to more complex problems, it lacks reliability across runs, demonstrating high potential but inconsistent execution.

- Qwen2.5-Coder-32B-Instruct underperforms expectations, especially considering its strong code generation capabilities. Analysis of its trajectories reveals problems with instruction following; the model frequently hallucinated environment responses or enters loops of formatting errors, ultimately failing without producing a meaningful solution attempt.

- Qwen3 models perform similarly with or without think mode enabled – in some cases, the no-think variant even slightly surpasses the think version. This suggests the base model's capabilities are sufficiently strong for deliberate planning to provide no measurable advantage. The nearly identical pass@5 scores further indicate that the model's problem-solving efficiency remains consistent even without explicit reasoning mechanisms

- DeepSeek models demonstrate the strongest performance among open-source models across both SWE-rebench subsets and the SWE-bench Verified benchmark. Notably, both the December and March releases of DeepSeek-V3 consistently outperform other open models in resolution rate and pass@5, highlighting their robustness to changes in task distribution. Interestingly, while both versions perform similarly on SWE-rebench, their scores diverge on SWE-bench Verified, which may suggest potential contamination effects on the older benchmark.

For evaluation details and experimental setup, see Appendix J.

# 4   Discussion and limitations

Our automated pipeline and the resulting SWE-rebench dataset are designed to address the lack of large-scale, real-world tasks for agent-based training, and the need for up-to-date benchmarks that remain free from data contamination. By automating the extraction and validation of executable tasks, we enable broad coverage and continual supply of fresh data. However, the emphasis on scalability introduces trade-offs, particularly a reduced ability to manually curate and verify the quality and clarity of each individual task.

Extracting consistently high-quality, verifiable SWE tasks from diverse real-world GitHub repositories (Section 2) is an inherently imperfect process. While our multi-stage filtering, refinements to existing methodologies and automated dependency installation are designed for robustness at scale, they rely on heuristics and LLM-driven interpretations. For instance, our LLM-based approach to generating installation instructions from repository files (Qwen2.5-72B-Instruct, Section 2.2), while far more scalable than manual methods, was validated on a limited set of 18 repositories for prompt engineering and may not capture every project's subtleties. Similarly, the automated task quality assessment (Section 2.4), where an LLM is fine-tuned on SWE-bench Verified task labels to predict complexity and relevance, serves as a valuable scalable proxy but cannot fully replicate nuanced human judgment, thus, containing errors and decreasing quality of the datasets.

Finally, while our benchmark is intended to support transparency and standardization in evaluating SWE agents, it may also accelerate the development of increasingly autonomous AI systems in software engineering. This progress brings potential risks, such as overreliance on AI-generated code or misuse of automated agents for introducing vulnerabilities. We believe that fostering openness, decontaminated evaluations, and rigorous benchmarking practices helps mitigate these concerns and contributes to responsible advancement of the field.

We outline following main limitations of our work:

- **Automated task quality assessment:** While we employ automated quality assessment, the fully automated pipeline may result in some tasks being imperfectly described or unsolvable solely from the issue. This can lead to lower absolute success rates compared to manually curated benchmarks.

- **Limited language diversity:** The initial release of SWE-rebench is limited exclusively to Python, restricting its immediate applicability to other language ecosystems. Although the underlying pipeline is language-agnostic, adapting it to support other languages (such as Go, Java, Rust, JavaScript/TypeScript) requires implementing language-specific components, which is planned for future work.

# 5   Conclusion and future work

We have introduced a novel, fully automated pipeline for the continuous collection of software engineering tasks from open-source repositories. This pipeline provides a scalable and reliable source of fresh, decontaminated data, particularly suitable for training and evaluating LLM-based agents. It serves as the foundation for SWE-rebench public datasets and a continuously updated benchmark designed for robust and transparent evaluation of agent performance in realistic SWE scenarios. By shifting the paradigm towards automated data collection and dynamic benchmarking, we address critical limitations of existing static benchmarks and thereby facilitate more rapid and open progress in the field of AI for software engineering. Our future work will concentrate on several key areas:

- **Expanding data coverage and volume:** We aim to significantly increase the dataset volume by extending our collection methodology from tasks strictly tied to GitHub issues to a broader set of code changes represented by arbitrary pull requests.

- **Improving task filtering pipeline:** To improve the overall quality of extracted tasks, we aim to refine our filtering heuristics used in the pipeline.

- **Support for new programming languages:** Applying the same core methodology, we plan to collect datasets for projects in other popular languages (e.g., JavaScript, Java, C++), thereby expanding the linguistic and technological diversity of SWE-rebench.

- **Keeping SWE-rebench benchmark up-to-date:** Maintaining evaluation process on fresh tasks for the existing models, evaluating a wider range of LLMs and sharing detailed performance analyses with the community.

We believe that our automated data collection pipeline and the SWE-rebench benchmark provide a vital foundation for developing and assessing the next generation of LLM-based agents for real-world software engineering challenges.

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

# A  Related work

**Benchmarks for software engineering**   With the progress of LLM capabilities, there has been a growing need for benchmarks to evaluating their coding performance. While HumanEval [Chen et al., 2021], MBPP [Austin et al., 2021], and APPS [Hendrycks et al., 2021] have proven valuable for standalone function-level evaluation, their utility has largely saturated for state-of-the-art models. Consequently, LiveCodeBench [Jain et al., 2024] has emerged, which is continuously updated to reflect the evolving capabilities of LLMs. However, these benchmarks often differ significantly from how LLMs are applied in real-world development workflows, which typically involve repository-level context and interaction. SWE-bench [Jimenez et al., 2024] introduced execution-based, repository-level tasks sourced from real-world repositories, offering a more realistic evaluation paradigm. A challenge with SWE-bench is that many new LLMs were released after its construction, raising concerns about its continued relevance and potential contamination. Several follow-up works, such as SWE-bench+ [Aleithan et al., 2024] and Multi-SWE-bench [Zan et al., 2025], address specific shortcomings like post-cutoff filtering and multilingual annotation. Nevertheless, all these extensions fundamentally rely on a manually curated construction process. This manual element complicates timely updates and makes it difficult to keep the benchmark consistently aligned with the rapidly advancing capabilities of LLMs. In contrast, SWE-rebench automates the entire process of task creation from real-world repositories at scale. This automation enables a new class of dataset and benchmark: one that is task-rich, diverse, regularly updated, and suitable for both training and evaluation without the bottlenecks of manual curation.

**Training datasets for code generation**   Most prior work on collecting data to improve LLMs for code has focused on supervised fine-tuning [Muennighoff et al., 2024, Luo et al., 2023, Liu et al., 2024], often by leveraging samples from more powerful proprietary models. On the other hand, the remarkable success of large-scale reinforcement learning (RL) in math reasoning tasks [DeepSeek-AI, 2025, Seed et al., 2025, OpenThoughts, 2025] implies that it should also excel in software engineering, as both domains involve rich, multi-step workflows with potentially verifiable outcomes. However, leveraging RL effectively in this setting demands gathering a large collection of real-world, interactive SWE tasks which is challenging, as each task requires the construction of a suitable and reliable execution environment. Some works attempt to gather such tasks by artificially injecting bugs into code [Yang et al., 2025a] or generating synthetic tests [Xie et al., 2025]. While these approaches can generate data at scale, they may not fully capture the nuances and complexities of real-world software issues. The original SWE-bench dataset and its extensions for training [Pan et al., 2024b] provide valuable real-world tasks, but still suffer from limited diversity and scale due to their reliance on manual construction. In contrast, SWE-rebench reflects real-world development processes and preserves high task quality through its grounding in real repositories, while achieving significantly greater scale and diversity via its automated construction pipeline. This makes it particularly well-suited for training agents through an automated construction pipeline. Indeed, the value of this approach is demonstrated by research successfully applying reinforcement learning to train software engineering agents [Luo et al., 2025, Cao et al., 2025], including work that directly utilizes our SWE-rebench dataset [Yang et al., 2025b].

# B  Automated dependency installation

As a reminder, the goal of automated dependency installation is to generate correct environment setup instructions for each task instance without using manual curation. The process is entirely LLM-driven and consists of the following sub-stages:

1. **Identifying relevant files:** An LLM is asked to analyze the list of repository files and select those most likely to contain installation instructions, such as `README.md`, `setup.py`, or `requirements.txt`.

2. **Extracting installation recipe:** Based on the content of files found in the previous stage, the model generates a structured JSON recipe specifying how to install dependencies and run tests for the repository.

3. **Updating the recipe (if needed):** If the recipe fails, another LLM call is made to analyze the failure and refine the recipe.

In the following subsections, we show the exact prompts used at each of these stages, along with examples of LLM-generated outputs. These materials are intended to provide transparency and facilitate reproducibility of our environment setup pipeline.

## B.1 Identifying installation-related files

The automated generation of installation recipes, a key part of our agentless approach described in Section 2.2, begins with identifying repository files that are likely to contain setup instructions. To achieve this, an LLM is provided with a list of all files present in the repository. The LLM's task is to analyze this list and return a curated subset of file paths deemed most relevant for extracting installation, dependency, and testing information. The prompt below details the instructions given to the LLM for this file identification step. The output of this step (a JSON array of file paths) subsequently serves as input for the next LLM call, which extracts the actual installation commands.

Listing 1: LLM prompt for identifying installation-related files

```
You are tasked with identifying files that likely contain installation
    instructions for a GitHub repository.

Repository: {{ repo_name }}

Below is a list of files in the repository that may be helpful for
    understanding the installation and setup process:

{{ list_of_files }}

Please analyze this list and identify the files that are most likely to
    contain information about:
  * Installation instructions
  * Dependencies or requirements
  * Setup procedures
  * Development environment configuration
  * Testing setup

Think step by step:
  * Identify README files, as they often contain installation instructions.
  * Look for setup.py, pyproject.toml, requirements.txt, environment.yml.
  * Consider files in directories like \texttt{docs} that might contain
      installation guides.
  * Look for test configuration files that might help understand how to run
       tests.
  * Consider \textbf{only} files from the list provided above.
  * Prioritize files in the repository root (top-level directory).
  * Only include files from subdirectories if they are clearly relevant to
      installation or setup.

Return **only** a JSON array containing the paths to the most relevant
    files for installation and setup. Include only files that are directly
    relevant to the tasks above. Sort the files from most to least relevant
     and limit your response to no more than 10 files, preferring fewer
    files that are truly essential.

For example:
[
    "README.md",
    "setup.py",
    "requirements.txt"
]
```

## B.2  Extracting installation recipe

Following the identification of relevant files, the next step in our automated pipeline (Section 2.2) is to extract a concrete installation recipe. For this, the LLM receives the concatenated content of the previously selected relevant files. The primary task of the LLM is to analyze this content and synthesize a structured JSON object, *installation recipe*, which encapsulates all necessary commands and configurations to set up the project environment, install dependencies, and execute tests. The prompt presented below guides the LLM through this extraction process, emphasizing the expected JSON structure and providing context on how the generated recipe will be utilized by downstream automation scripts.

Listing 2: LLM prompt for extracting installation recipe

```
You are tasked with extracting detailed installation instructions from the
    following repository files. Repository: {{ repo_name }}

Please analyze the content of these files and extract comprehensive
    installation instructions: {{ rendered[:100000] }}

First, think step by step. After your analysis, return your findings in the
    following JSON format:

{
  "python": "3.9",
  "packages": "requirements.txt",
  "install": "pip install -e .[dev]",
  "test_cmd": "pytest --no-header -rA --tb=line --color=no -p no:
      cacheprovider -W ignore::DeprecationWarning",
  "pre_install": ["apt-get update", "apt-get install -y gcc"],
  "reqs_path": ["requirements/base.txt"],
  "env_yml_path": ["environment.yml"],
  "pip_packages": ["numpy>=1.16.0", "pandas>=1.0.0"]
}

Here is how this JSON will be used:

```bash
git clone <repo_url> repo
cd repo
git checkout <base_sha>
bash <pre_install>
conda create -n <repo> python=<python> -y
conda activate <repo>
if <packages> == requirements.txt; then
  for path in <reqs_path>:
      pip install -r $path
elif <packages> == environment.yml; then
  for path in <env_yml_path>:
      conda env update -f $path
else:
  pip install <packages>
pip install <pip_packages>
bash <install>
bash <test_cmd>
```

**IMPORTANT:**
  * For the "install" field, always use local install commands like pip
      install -e .[dev]
```

```
    * Do NOT include packages in pip_packages that will be installed by pip
        install -e .
    * Include only explicitly needed packages in pip_packages.
    * reqs_path and env_yml_path must match filenames from the provided files
        (e.g., [File: filename]).
    * If "packages" is requirements.txt, you must provide at least one
        reqs_path.
    * Add relevant test frameworks to pip_packages (e.g., pytest, nose).
    * Use -y in all conda commands.
    * Prefer direct and specific pytest commands over general wrappers.
    * Avoid test commands with placeholders like {test_name}.
    * If a Makefile runs tests, extract the actual test command (e.g., pytest)
        .

You must ensure the final JSON includes required fields (, install,
    test_cmd), and optionally packages, pre_install, reqs_path,
    env_yml_path, pip_packages if relevant.

Base your reasoning on all provided files and return it followed by the
    JSON output in a Markdown code block (without comments).
```

### B.3 Example of LLM-generated installation recipe

Below is an illustrative example of an output from the installation recipe extraction stage that includes both the reasoning and the final JSON.

Listing 3: LLM-generated installation recipe

```
1. Python version. The setup.py file specifies that the project requires
    Python 3.6 or greater. The README.rst and setup.py files also indicate
    that the project supports Python 3.6, 3.7, 3.8, 3.9, and 3.10. For a
    safe middle-ground, Python 3.9 is chosen.

2. Installation Method. The README.rst and DEVELOPMENT.md files both
    mention using pip to install the project. Specifically, the development
     installation command is pip install -e .[dev]

3. Dependencies. The setup.py file lists flake8 >= 3.0.0 and attrs >=
    19.2.0 as dependencies. The DEVELOPMENT.md file also mentions
    additional development dependencies: coverage, hypothesis, hypothesmith
     >= 0.2, and pre-commit. These are included in the pip_packages field.

4. Test command. The README.rst and DEVELOPMENT.md files both mention
    running tests with coverage run tests/test_bugbear.py. However, for
    simplicity and to follow the guidelines, the test command is simplified
     to pytest tests/test_bugbear.py.

5. Pre-install steps. No system-level pre-install steps are mentioned in
    the provided files.

6. Additional packages. The pytest package is needed for running tests, and
     it is not explicitly listed in the project's dependencies. Therefore,
    it is added to the pip_packages field.

{
  "python": "3.9",
  "install": "pip install -e .[dev]",
  "test_cmd": "pytest tests/test_bugbear.py",
  "pip_packages": [
```

```
      "pytest",
      "coverage",
      "hypothesis",
      "hypothesmith>=0.2",
      "pre-commit"
  ]
}
```

## B.4   Updating installation recipe

As described in Section 2.2, our agentless installation recipe generation includes an iterative refine-
ment step. If the initial attempt to set up an environment using an LLM-generated recipe fails, we
trigger a correction mechanism. This mechanism involves a subsequent call to the LLM, providing it
with the original (failed) installation recipe, the error logs produced during the failed setup or test
execution, as well as the contents of relevant files. The LLM's objective is then to diagnose the errors
based on the logs and propose modifications to the recipe to resolve these issues. The prompt detailed
below instructs the LLM on how to perform this error analysis and generate an updated, corrected
installation recipe in the same structured JSON format.

Listing 4: LLM prompt for updating installation recipe

```
You are an expert in fixing software installation and testing issues.
    Analyze the installation logs provided and update the installation
    configuration to fix any errors.

Install config fields:

{
  "python": "3.9",
  "packages": "requirements.txt",
  "install": "pip install -e .[dev]",
  "test_cmd": "pytest --no-header -rA --tb=line --color=no -p no:
      cacheprovider -W ignore::DeprecationWarning",
  "pre_install": ["apt-get update", "apt-get install -y gcc"],
  "reqs_path": ["requirements/base.txt"],
  "env_yml_path": ["environment.yml"],
  "pip_packages": ["numpy>=1.16.0", "pandas>=1.0.0"]
}

Current installation configuration:
{{ install_config }}

Error logs from installation/testing (last relevant lines): {{ cut_logs }}

Your task:
 * Identify the root causes of the errors in the logs.
 * Modify the installation configuration to address these issues.
 * You might need to:
   - Add missing dependencies
   - Fix command syntax errors
   - Change installation order
   - Add environment variables
   - Modify test commands

First, provide brief reasoning (<100 words) about what's causing the errors
    and your planned fixes.

Then return the complete updated installation configuration as a valid JSON
    object.
```

## B.5 Failure modes during execution validation and mitigation strategies

During execution validation, we identified several recurring failure modes that prevent otherwise valid tasks from being included in the dataset:

- **Flaky tests with external API calls or floating-point precision issues:** Many failures were attributed to tests involving external API calls to intermittently unavailable endpoints or tests asserting exact equality of floating-point numbers. These failures are non-deterministic and do not reliably reflect task quality. Since we cannot depend on such tests for consistent verification, we run tests multiple times and exclude instances that fail in at least one run.

- **Lack of interactivity during installation:** Some projects, particularly older ones, require an interactive approach during the installation process, where dependencies must be incrementally installed or configurations adjusted based on intermediate error messages. Such iterative troubleshooting cannot be reliably accomplished by an agent with a small number of predefined steps, leading to installation failures even when the underlying task is valid.

- **Unpinned and obsolete dependencies:** Environment setup frequently failed for older repositories where README instructions were accurate but dependency versions were not pinned or were no longer publicly available. To mitigate this, we freeze all dependency versions after successful setup and maintain internal package mirrors for common dependencies, significantly reducing setup failures on retry.

These failure modes account for most validation rejections, and addressing them—particularly through more interactive installation mechanisms or advanced dependency resolution—represents a key opportunity for improving the pipeline's yield.

## C Evaluation of automated installation recipe generation

To assess the efficacy of our automated installation recipe generation and to select effective prompts, we conducted a validation study. We utilized a curated set of 18 task instances, each from a distinct repository within the original SWE-bench dataset. These instances were chosen because their environment installation instructions were originally manually crafted, providing a reliable baseline. For each instance, we attempted to automatically generate an installation recipe, set up the corresponding environment, and execute its tests.

The performance of the agentless LLM-based recipe generation (with varying numbers of candidate generations) was compared against an interactive agent designed for the same purpose. The results are summarized in Table 3. Some unsuccessful configuration attempts are due to limitations in our automated log parsing during verification, especially for repositories with custom test frameworks such as Django. Our generic parsers (e.g., for pytest) may fail to extract test results from highly customised test outputs. Nevertheless, this small-scale validation served as a sanity check and informed our selection of prompts and the number of candidate generations for the agentless approach in the main pipeline. The list of specific SWE-bench instances used for this validation can be found in Appendix C.1.

Table 3: Comparison of installation method success rates on 18 SWE-bench tasks.

| Approach | Candidates/Trials | Successfully Configured |
|---|---|---|
| Agentless | 1 | 6 out of 18 |
| Agentless | 3 | 8 out of 18 |
| Agentless | 10 | 9 out of 18 |
| Agent-based | 1 | 8 out of 18 |

## C.1 List of instances to validate automatic installation instruction extraction

- `astropy__astropy-12907`
- `django__django-15315`
- `marshmallow-code__marshmallow-1164`

- `matplotlib__matplotlib-20826`
- `mwaskom__seaborn-3069`
- `pallets__flask-5014`
- `psf__requests-2317`
- `pvlib__pvlib-python-1160`
- `pydata__xarray-6744`
- `pydicom__pydicom-996`
- `pylint-dev__astroid-1741`
- `pylint-dev__pylint-4604`
- `pytest-dev__pytest-7982`
- `pyvista__pyvista-4853`
- `scikit-learn__scikit-learn-13142`
- `sphinx-doc__sphinx-8721`
- `sqlfluff__sqlfluff-4764`
- `sympy__sympy-22080`

## D    Permissive licenses included in data collection

Our data collection process exclusively targeted repositories under permissive open-source licenses. We identified repositories matching common permissive licenses using their SPDX identifiers where available. For repositories with licenses not matching this predefined list, or those with custom license text (categorized as "Other"), a manual review was conducted to confirm that their terms permit the use cases relevant to our work. The primary permissive license types included were:

- MIT
- Apache-2.0
- BSD-4-Clause
- BSD-3-Clause
- BSD-2-Clause
- ISC
- CC0-1.0
- ZPL-2.1
- Other (manually verified permissive licenses)

## E    Example of a task instance with annotations

Below is an example of a single task instance from the SWE-rebench dataset, illustrating the structure and some of the key annotations collected or generated by our pipeline. For brevity, some lengthy fields like full file contents or complete dependency lists were truncated or summarized. The instance is presented in the form of a Python dictionary.

Listing 5: SWE-rebench task instance contents example

```
{
  'instance_id': 'tarohi24__typedflow-68',
  'repo': 'tarohi24/typedflow',
  'base_commit': '2127e74314d2b97d596cfc12ed8fb257bb688d6f',
  'version': '1.0',
  'created_at': '2019-12-10 15:26:34',
```

```
      'problem_statement': "The new syntax doesn't work\nIt doesn't accept args
          in the correct way. For instance, life of cache tables are never
          incremented.",
      'patch': """diff --git a/typedflow/nodes/base.py b/typedflow/nodes/base.
          py
index ece0895..b9853f9 100644
--- a/typedflow/nodes/base.py
+++ b/typedflow/nodes/base.py
@@ -113,7 +113,8 @@ class ConsumerNode:
          None
          """
          assert len(self.precs) == 0, 'Some arguments have been already set'
- self.precs: Dict[str, ProviderNode] = args
+ for name, prec in args.items():
+ self.set_upstream_node(name, prec)
          return self

""",
      'test_patch': """diff --git a/typedflow/tests/flow/test_flow.py b/
          typedflow/tests/flow/test_flow.py
index aa31917..7682475 100644
--- a/typedflow/tests/flow/test_flow.py
+++ b/typedflow/tests/flow/test_flow.py
@@ -209,3 +209,4 @@ def
    test_declare_inputs_when_definition_with_multiple_args():
      node_dump = DumpNode(dump)({'a': node_task})
      flow = Flow([node_dump, ])
      flow.typecheck()
+ assert node_task.cache_table.life == 1
""",
      'meta': {
        'commit_name': 'head_commit',
        'num_modified_files': 1,
        'llm_score': {'issue_text_score': 3, 'test_score': 1, 'difficulty_score
            ': 1}
        # Other meta fields like 'failed_lite_validators', 'has_test_patch', '
            is_lite'
      },
      'install_config': {
        'python': '3.8',
        'packages': 'requirements.txt',
        'reqs_path': ['requirements.txt'],
        'install': 'pip install -e .',
        'pip_packages': ['pytest', 'pytest-runner', ...], // Truncated for
            brevity
        'pre_install': ['apt-get update', 'apt-get install -y gcc build-
            essential ...'], // Truncated
        'test_cmd': 'pytest --no-header -rA --tb=line --color=no -p no:
            cacheprovider ...' # Truncated
        # Other install_config fields like 'env_vars', 'env_yml_path', '
            log_parser'
      },
      'FAIL_TO_PASS': ['typedflow/tests/flow/test_flow.py::
          test_declare_inputs_when_definition_with_multiple_args'],
      # Other test outcome fields like 'FAIL_TO_FAIL', 'PASS_TO_PASS', '
          PASS_TO_FAIL'
      'license_name': 'MIT License'
```

```
  # Fields like 'requirements', 'environment', 'hints_text', 'total_len'
      also exist.
}
```

This example showcases the core components of a task: the problem description, the code changes (`patch` and `test_patch`), and associated metadata including LLM-generated quality scores and installation configurations. The full dataset contains more extensive information about each instance.

# F  Comparison of models for automatic task quality assessment

As mentioned in Section 2.4, we fine-tuned a Qwen 2.5-72B-Instruct model using human annotations from SWE-bench Verified to predict three quality assessment labels: **Test Patch Correctness**, **Task Complexity**, and **Issue Clarity**. Table 4 provides a detailed classification quality report, comparing the performance of our fine-tuned model against the vanilla Qwen 2.5-72B-Instruct model on the validation set.

To validate the utility of file count as a difficulty heuristic, we conducted an empirical analysis of DEEPSEEK-V3-0324 performance across SWE-rebench leaderboard tasks (January–July 2025), segmented by the number of files changed: (i) 1 file: $28.6\% \pm 0.8\%$ ($n = 201$); (ii) 2 files: $20.6\% \pm 0.4\%$ ($n = 144$); (iii) $\geq 3$ files: $17.5\% \pm 2.2\%$ ($n = 64$). These results confirm that performance decreases as the number of changed files increases, supporting this metric as a coarse difficulty proxy. However, as noted above, task difficulty is better captured via direct quality assessment rather than file count alone: a multi-file change can be simple (e.g., repeated parameter updates), whereas a single-file change can be challenging due to unclear issue descriptions or inadequate tests.

Table 4: Classification report for task label prediction, comparing the vanilla Qwen-2.5-72B-Instruct model with its fine-tuned counterpart.

| Label | | Instruct | | | Fine-tuned | | | Support |
|---|---|---|---|---|---|---|---|---|
| | | Prec | Rec | F1 | Prec | Rec | F1 | |
| Test Patch | low-score | 0.59 | 0.97 | 0.73 | 0.66 | 0.85 | 0.74 | 233 |
| Correctness | high-score | 0.76 | 0.12 | 0.21 | 0.69 | 0.42 | 0.52 | 180 |
| | accuracy | | | 0.60 | | | 0.67 | 413 |
| | macro avg | 0.67 | 0.55 | 0.47 | 0.67 | 0.64 | 0.63 | 413 |
| | weighted avg | 0.66 | 0.60 | 0.51 | 0.67 | 0.67 | 0.65 | 413 |
| Task | low-score | 0.90 | 0.64 | 0.75 | 0.90 | 0.85 | 0.87 | 313 |
| Complexity | high-score | 0.41 | 0.78 | 0.54 | 0.60 | 0.71 | 0.65 | 100 |
| | accuracy | | | 0.68 | | | 0.81 | 413 |
| | macro avg | 0.66 | 0.71 | 0.64 | 0.75 | 0.78 | 0.76 | 413 |
| | weighted avg | 0.78 | 0.68 | 0.70 | 0.83 | 0.81 | 0.82 | 413 |
| Issue | low-score | 0.83 | 0.94 | 0.88 | 0.82 | 0.94 | 0.88 | 329 |
| Clarity | high-score | 0.51 | 0.26 | 0.35 | 0.47 | 0.20 | 0.28 | 84 |
| | accuracy | | | 0.80 | | | 0.79 | 413 |
| | macro avg | 0.67 | 0.60 | 0.61 | 0.65 | 0.57 | 0.58 | 413 |
| | weighted avg | 0.77 | 0.80 | 0.77 | 0.75 | 0.79 | 0.76 | 413 |

# G  Refinements to the original SWE-bench methodology

Our goal is to create a dataset suitable not just for evaluation, but also for reinforcement learning. Ideally, its tasks should remain unsolved only due to the agent's inherent limitations, not due to faulty tests or incorrect problem specifications. This is why we enhanced mechanisms ensuring task validity compared to those used to build SWE-bench.

- **Patch generation from git history:** In the original SWE-bench, the diff between `base_commit` (where the PR branch starts) and `merge_commit` (the PR is merged into the main branch) forms the solution and test patches. However, intervening merges into the main branch can introduce unrelated changes into this diff, potentially invalidating tasks by including tests for functionality that is external with respect to the PR. As an example, `sympy__sympy-14821` was not included in SWE-bench Verified for this reason. To mitigate this problem, we prioritize diffing `head_commit` (the last commit in the PR branch) against `base_commit` to generate patches, as this isolates changes made directly within the branch. If `head_commit` is unavailable (e.g., due to branch deletion after merge), we fall back to `merge_commit` and record this choice in the task metadata.

- **Test directive generation:** SWE-bench generates test directives (commands to run specific tests for verification) from the test patch using regular expressions. This can erroneously include deleted test files, leading to invalid commands. Our refinement filters out deleted files from test directives.

- **AttributeError/ImportError checks:** Tasks where tests check for new, not-yet-existing attributes or imports can be problematic, as the agent must guess exact signatures when implementing a solution. While often detectable, such errors can be masked if tests are being run with reduced traceback verbosity (e.g., Pytest's `tb=no` flag, used in SWE-bench for some repositories). We run tests with full error output (`tb=line`) to accurately identify these cases. This information is recorded in task metadata, allowing for optional filtering, as some instances (e.g., fixing an import error) are valid.

- **Dependency pinning:** Python projects often lack explicitly pinned dependency versions. Defaulting to the latest versions during installation can lead to test failures due to package compatibility changes over time. To ensure reproducibility, we freeze and record all dependency versions (using `pip freeze` and `conda env export`) after the first successful environment setup for a task (or a task group sharing the same environment). These pinned versions are then reused for subsequent rebuilds.

## H   Filtering criteria for the SWE-rebench benchmark subset

The SWE-rebench benchmark subset, used for evaluating LLM-based agents as described in Section 3, is curated from the larger SWE-rebench dataset by applying a series of specific filtering criteria. These filters are designed to select tasks that are well-defined, of manageable complexity for current models, and ensure a consistent evaluation environment. The following conditions must be met for a task instance to be included in the benchmark subset:

- **Clean test execution:** Test execution logs prior to applying any solution patch must be free of critical errors such as `AttributeError` or `ImportError`.

- **Code modification scope:** The number of files modified by the solution patch must be no more than 3.

- **Patch size:** The total number of words in the solution patch must not exceed 500.

- **Problem statement length:** The problem statement (github issue description) must contain between 16 and 1000 words (inclusive).

- **Problem statement language:** The problem statement must be in English.

- **Recency:** The github issue the task is based on must have been created in the year 2025.

- **Assessed difficulty:** The LLM-assessed difficulty label for the task must be less than 3, indicating low to moderate complexity.

- **Test case count:** The number of tests that transition from a failing to a passing state (fail-to-pass tests) must be 50 or fewer.

## I   System prompt for agent runs

Listing 6: System prompt for LLM evaluation runs on SWE-rebench leaderboard tasks

```
# SETTING
```

You are an autonomous programming agent. Your goal is to resolve the issue
    given to you.
You are given access to a terminal environment with some special tools to
    make your job easier.
You must use the terminal to gain information about the codebase, find or
    modify the relevant files in order to resolve the issue.
In this environment, all standard unix commands (e.g. grep, sed, echo etc.)
     will be available to you.
However, the environment does NOT support interactive session commands
    that expect user input (e.g. vim), so please do not invoke them, it
    will result in an error
You can however create python scripts and run them, this is very useful to
     reproduce errors or test something.
If some packages are missing, you can install them using an appropriate
    package manager (e.g. pip, apt, etc.).
Do not ask any questions to the environment, it's an automated system that
     can only execute your commands.
When you are satisfied with the changes you made, you should explicitly
    submit them using a special command. This will terminate your session.

# SPECIAL TOOLS

In addition to standard unix commands you can use special tools described
    below.
Please note that some of these commands work with the currently open file,
     so pay attention to what file is open.

Usage: create [OPTIONS] FILENAME
  Creates and opens a new filename with the given name.

Usage: edit [OPTIONS] LINE_RANGE [REPLACEMENT_TEXT]
  Replaces lines in LINE_RANGE=<start_line>:<end_line> (inclusive) with the
  given text in the currently open or specified file. The REPLACEMENT_TEXT
  will be used as provided including all whitespaces, so make sure your
  indentation is correct.
  To input multiple lines into REPLACEMENT_TEXT, you may use the following
  syntax:
  ```
  edit 1:1 << 'EOF'
  Line1
  Line2
  EOF
  ```
  You can also provide the file to edit via `--file` option.
  ```
  edit --file path/to/file 1:1 "Your Replacement Text Here"
  ```
  Please note that THIS COMMAND REQUIRES PROPER INDENTATION. If you'd like
      to
  add the line ' print(x)' you must fully write that out, with all
  those spaces before the print statement!
Options:
  --file PATH The file to edit. (If not provided, edits the currently open
              file)

Usage: goto [OPTIONS] LINE_NUMBER
  Navigates the current window to a given line in the currently open file.

```
Usage: open [OPTIONS] [FILE] [LINE_NUMBER]
  Opens the file at the given path in the editor. If file is not specified,
  the last open file will be reopened. If line_number is provided, the
      current
  window will move to show that line.

Usage: replace [OPTIONS] SEARCH REPLACE
  Replaces a given string with another string in the currently open file.
Options:
  --replace-all Replace all occurrences of the SEARCH text.

Usage: scroll_down [OPTIONS]
  Scroll down the window in the currently open file and output its contents
      .

Usage: scroll_up [OPTIONS]
  Scroll up the window in the currently open file and output its contents.

Usage: search_file [OPTIONS] SEARCH_TERM [FILE]
  Searches for SEARCH_TERM in file. If FILE is not provided, searches in
      the currently open file.

Usage: submit [OPTIONS]
  Submits your current code and terminates the session.

# ENVIRONMENT RESPONSE

At the very beginning the environment will provide you with an issue
    description. In response to every command that you invoke,
the environment will give you the output of the command or an error
    message followed by a shell prompt.
The shell prompt will be formatted as follows:
```
(Current directory: <current_dir>, current file: <current_file>) bash-$
```
so that you always know what the current directory is and what file is
    currently open.

# YOUR RESPONSE

Your response should consist of two parts: reasoning (arbitrary text) and
    command (surrounded by triple ticks and a special 'command' keyword).
Your response should always include A SINGLE reasoning and A SINGLE
    command as in the following examples:

<response example>
First I'll start by using ls to see what files are in the current
    directory. I'll look at all files including hidden ones.
```command
ls -a
```
</response example>

<response example>
Let's search the file `models.py` for the UserEntity class definition.
```command
search_file "class UserEntity" models.py
```
```

</response example>

Everything you include in the reasoning will be made available to you when
    generating further commands.
If you'd like to issue two command blocks in a single response, PLEASE DO
    NOT DO THAT! THIS WILL RESULT IN AN ERROR.

# HANDLING TESTS

* You can run existing tests to validate the changes you made or make sure
    you didn't break anything.
* If missing packages or some environment misconfiguration is preventing
    you from running the tests, you can install missing packages or fix the
    environment.
* However UNDER NO CIRCUMSTANCES should you modify existing tests or add
    new tests to the repository.
  This will lead to an error in the system that evaluates your performance.
       Instead, you can just create a temporary script, use it to test
       changes and remove it before submitting.
* If existing tests break because they need to be updated to reflect the
    changes you made, just ignore it. Evaluation system will not take it
    into account.
* However if existing tests are broken because your fix is incorrect, you
    should fix your code and make sure all tests pass before submitting the
    change.

# USEFUL ADVICE

* As a first step, it might be a good idea to explore the repository to
    familiarize yourself with its structure.
* You should also come up with a rough plan of how to resolve the issue
    and put it into your reasoning.
* If the issue description reports some error, create a script to
    reproduce the error and run it to confirm the error. THIS IS USUALLY A
    VERY GOOD FIRST STEP!
* Edit the source code of the repo to resolve the issue
* Rerun your reproduce script and confirm that the error is fixed! THIS IS
    QUITE IMPORTANT!
* Think about edge cases and make sure your fix handles them as well.
* Make sure your solution is general enough and not hardcoded to the
    specific cases reported in the issue description.
* It might be a good idea to ensure that existing tests in the repository
    pass before submitting the change. Otherwise it's easy to break
    existing functionality.

# DEMONSTRATION

Here is a very simple demonstration of how agent can interact with the
    environment using the provided interface.

<demonstration><environment>
# ISSUE DESCRIPTION

Here is a script that is supposed to print out first 10 prime numbers, but
    it seems to have a bug. Can you fix it?

```
def is_prime(n):
    if n <= 1:
```

```
        return False
    for i in range(2, int(n**0.5)):
        if n % i == 0:
            return False
    return True

primes = []
num = 2
while len(primes) < 10:
    if is_prime(num):
        primes.append(num)
    num += 1

print("First 10 prime numbers:", primes)
```

# ADDITIONAL ADVICE

Since you are given a git repository, you can use git commands to simplify
     your work.
For example, if you made a mistake and want to revert your changes, you
    can use `git checkout <file>` to restore the file to its original state
    .
You can also reset all changes in the repository using `git reset --hard`
    command.
Additionally, you can use `git stash` and `git stash pop` to temporarily
    save your changes and restore them later.
Finally, you can see the changes that you've made (which will be submitted
     when you call `submit`) using `git status` or `git diff` commands.
However you don't need to use `git add` or `git commit` before submitting
    your changes. In fact,
`submit` only submits currently unstaged changes because it uses `git diff
    ` to get the changes that need to be submitted.

# CHECKLIST

Before submitting your solution, please go over the following checklist
    and make sure you've done everything:
- [ ] If an error was reported in the issue description, I have
    successfully reproduced it.
- [ ] If an error was reported in the issue description, I have confirmed
    that my fix resolves the error.
- [ ] I have thought about edge cases and made sure my fix handles them as
     well.
- [ ] I have run existing tests in the repository that might have been
    affected by the change I made and confirmed that they pass.
I want you to list every bullet from this checklist and write your
    reasoning for why you think you did it or didn't need to.

Repository has been uploaded and your shell is currently at the repository
     root. Time to solve the issue!

(Current directory: /project, current file: none) bash-$
</environment>
<agent>
Let me analyze the issue and fix the prime number generator script step by
     step.
```

First, let's create a Python script to reproduce the reported issue and
    verify the current behavior.

```command
create prime_numbers.py
```
</agent>
<environment>
[File: prime_numbers.py (1 lines total)]
1:

(Current directory: /project, current file: none) bash-$
</environment>
<agent>
Now I'll add the provided code to the file to test it and see what output
    it produces. The issue description mentions there's a bug, but doesn't
    specify what exactly is wrong. First we need to identify the issue.

```command
edit 1:1 << 'EOF'
def is_prime(n):
    if n <= 1:
        return False
    for i in range(2, int(n**0.5)):
        if n % i == 0:
            return False
    return True

primes = []
num = 2
while len(primes) < 10:
    if is_prime(num):
        primes.append(num)
    num += 1

print("First 10 prime numbers:", primes)
EOF
```
</agent>
<environment>
File updated. Please review the changes and make sure they are correct (
    correct indentation, no duplicate lines, etc). Edit the file again if
    necessary.
[File: /project/prime_numbers.py (16 lines total)]
1:def is_prime(n):
2: if n <= 1:
3: return False
4: for i in range(2, int(n**0.5)):
5: if n % i == 0:
6: return False
7: return True
8:
9:primes = []
10:num = 2
11:while len(primes) < 10:
12: if is_prime(num):
13: primes.append(num)
14: num += 1
15:

```
16:print("First 10 prime numbers:", primes)
```

(Current directory: /project, current file: none) bash-$
</environment>
<agent>
Now let's run the script to see what output it produces and identify the
    bug.

```command
python prime_numbers.py
```
</agent>
<environment>
First 10 prime numbers: [2, 3, 4, 5, 6, 7, 8, 9, 11, 13]

(Current directory: /project, current file: none) bash-$
</environment>
<agent>
Ah, I see the issue! The script incorrectly includes numbers 4, 6, 8, and
    9 in the output as "prime" numbers when they're actually not prime. The
     bug is in the `is_prime()` function - the range should go up to `int(n
    **0.5) + 1` because the upper bound in Python's range is exclusive. Let
    's fix this by modifying the range in the `is_prime()` function.

```command
edit 4:4 << 'EOF'
    for i in range(2, int(n**0.5) + 1):
EOF
```
</agent>
<environment>
File updated. Please review the changes and make sure they are correct (
    correct indentation, no duplicate lines, etc). Edit the file again if
    necessary.
[File: /project/prime_numbers.py (16 lines total)]
1:def is_prime(n):
2: if n <= 1:
3: return False
4: for i in range(2, int(n**0.5) + 1):
5: if n % i == 0:
6: return False
7: return True
8:
9:primes = []
10:num = 2
11:while len(primes) < 10:
12: if is_prime(num):
13: primes.append(num)
14: num += 1
15:
16:print("First 10 prime numbers:", primes)

(Current directory: /project, current file: none) bash-$
</environment>
<agent>
Now let's run the script again to verify that the fix works and produces
    the correct list of prime numbers.

```command
```
</agent>
```

```
python prime_numbers.py
```
</agent>
<environment>
First 10 prime numbers: [2, 3, 5, 7, 11, 13, 17, 19, 23, 29]

(Current directory: /project, current file: none) bash-$
</environment>
<agent>
The output now correctly shows the first 10 prime numbers. Let's go
    through the checklist:

1. Reproduced the error: Yes, we saw the script incorrectly included non-
    prime numbers in its output.
2. Confirmed the fix resolves the error: After modifying the range, the
    script now correctly lists only prime numbers.
3. Edge cases considered: The fix handles all cases correctly - numbers
    <=1 are non-prime, and the +1 ensures we check all potential divisors
    up to the square root.
4. Existing tests: Since this is a standalone script and there are no
    existing tests mentioned in the repository, this point isn't applicable
    .

The fix is complete and correct. We can now submit the changes.

```command
submit
```

</agent>
</demonstration>
```

## J   Evaluation details and experimental setup

To evaluate open-source models, we deployed them internally using the vLLM [Kwon et al., 2023] inference engine. Each model was served from 2 nodes equipped with 8×H200 GPUs each, and we used generation hyperparameters recommended by the respective model developers (e.g., temperature, p in top-p, max context length etc.). All evaluations were conducted using the same ReAct-style scaffolding and the system prompt described in Appendix I.

All evaluations were conducted using the same set of 294 SWE-rebench tasks. To reduce variance and enable pass@k computation, each model was run 5 times on the full benchmark using different random seeds. Among the evaluated models, DeepSeek-V3 was the largest and most resource-intensive; a single run over the full task set required approximately 7 hours.

As part of our analysis, we examined the overlap in successfully solved tasks across models to understand their relative capabilities and potential complementarity. Figure 2 visualizes the overlap of tasks solved at least once across selected models. While overall performance varies, the figure highlights that certain tasks are uniquely solved by specific models, suggesting complementary strengths.

### J.1   GPT-4.1 Trajectory Example

To better understand agent behavior on SWE-rebench tasks, we present a trajectory example that illustrates how a model attempts to resolve an issue—searching for relevant files, making edits, running tests, and reacting to feedback. For brevity, some of the observations, e.g., ones containing file contents, are truncated.

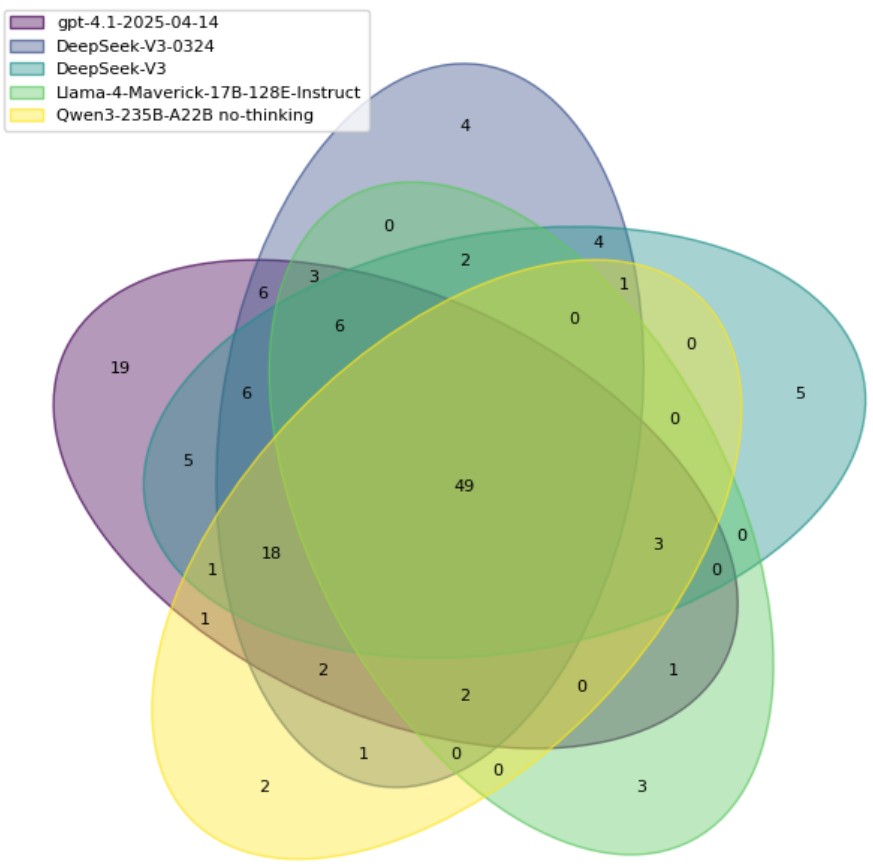

Figure 2: Overlap of solved tasks across selected models.

---

**System Prompt**

See Appendix I

---

**Environment Observation**

```
I've uploaded a python code repository, your shell is
    currently at the repository root. Consider the
    following PR description:

DESCRIPTION:
Network.plot_attribute
I find something wrong with:

```python
    if attribute[0].lower() == "z":
        y_label = "Z (Ohm)"
        y[x == 1.] = 1. + 1e-12 # solve numerical
            singularity
```

```
        y[x == -1.] = -1. + 1e-12 # solve numerical
            singularity
        y = self.z0[0].real * (1+y) / (1-y)
```

I think it should be:
```python
    if attribute[0].lower() == "z":
        y_label = "Z (Ohm)"
        y[y == 1.] = 1. + 1e-12 # solve numerical
            singularity
        y[y == -1.] = -1. + 1e-12 # solve numerical
            singularity
        y = self.z0[0].real * (1+y) / (1-y)
```

Am I right?

Can you help me implement the necessary changes to the
    repository so that the requirements specified in
    the <pr_description> are met? I've already taken
    care of all changes to any of the test files
    described in the <pr_description>. This means you
    DON'T have to modify the testing logic or any of
    the tests in any way!

Your task is to make the minimal changes to non-tests
    files in the {location} directory to ensure the <
    pr_description> is satisfied.

Follow these steps to resolve the issue:
1. As a first step, it might be a good idea to explore
    the repo to familiarize yourself with its structure.

2. If <pr_description> reports some error, create a
    script to reproduce the error and execute it with
    python <filename.py>, to confirm the error
3. Edit the sourcecode of the repo to resolve the issue
4. Rerun your reproduce script and confirm that the
    error is fixed!
5. Think about edgecases and make sure your fix handles
    them as well

Your thinking should be thorough and so it's fine if it
    's very long.
(Current directory: /scikit-rf)
bash-$
```

**Agent Output**

Let's start by searching for the plot_attribute method
in the repository to locate where the code exists
and verify the current implementation that is in
question.

```
grep -rn "def plot_attribute"
```

```
skrf/network.py:4830: def plot_attribute( self,

(Current directory: /scikit-rf)
bash-$
```

**Agent Output**

The plot_attribute function is located in skrf/network.
py at line 4830. Let's open the surrounding lines
to inspect the relevant section and verify the code
snippet described in the issue.

```
open skrf/network.py 4830
```

**Environment Observation**

```
[File: skrf/network.py (8517 lines total)]
...
(Current directory: /scikit-rf, Current file: /scikit-
    rf/skrf/network.py)
bash-$
```

**Agent Output**

We can see in the plot_attribute function, around line
4872, the code matches the initial suggestion:

```
if attribute[0].lower() == "z":
    y_label = "Z (Ohm)"
    y[x == 1.] = 1. + 1e-12 # solve numerical
        singularity
    y[x == -1.] = -1. + 1e-12 # solve numerical
        singularity}
    y = self.z0[0].real * (1+y) / (1-y)
```

The issue points out that y[x == 1.] should really be
y[y == 1.], as the intent is to fix elements of y
with possible singularity issues, not x.
The correction is:

```
y[y == 1.] = 1. + 1e-12
y[y == -1.] = -1. + 1e-12
```

Let's replace these two lines with the correct ones.

```
edit 4872:4873 << 'EOF'
           y[y == 1.] = 1. + 1e-12 # solve numerical
                singularity
           y[y == -1.] = -1. + 1e-12 # solve numerical
                singularity
EOF
```

File updated. Please review the changes and make sure
they are correct (correct indentation, no duplicate
lines, etc). Edit the file again if necessary.

[File: /scikit-rf/skrf/network.py (8517 lines total)]
...
(Current directory: /scikit-rf, Current file: /scikit-
     rf/skrf/network.py)
bash-$

**Agent Output**

The code has been updated as intended. Now, I need
to ensure that the fix behaves as expected and
doesn't cause any regressions. I'll check if there are
tests that might cover this situation and run them.
Let's look for a tests directory or test files
in the repository.

```
ls -l
```

**Environment Observation**

total 52
...
(Current directory: /scikit-rf, Current file: /scikit-
     rf/skrf/network.py)
bash-$

**Agent Output**

There is a skrf directory which is likely where the
main package code lives, but there is no obvious
tests directory here. Let's look inside the skrf
directory to see if the tests are located there.

```
ls -l skrf
```

**Agent Output**

There is a tests directory inside skrf. Let's list its contents to see what test files are available and determine if there is test coverage for plotting or the plot_attribute method.

```
ls -l skrf/tests
```

**Agent Output**

The skrf/tests directory contains several test files, including test_network.py and test_plotting.py, both of which are likely candidates for testing the (now-fixed) code. To ensure nothing is broken, let's run the tests in this directory.

I'll use pytest to run all tests in skrf/tests.

```
pytest skrf/tests
```

## J.2 Qwen2.5-32B Coder hallucination example

Below is a representative failure case for Qwen2.5-Coder-32B-Instruct. Despite its strong performance on various code generation benchmarks, this model repeatedly hallucinates environment responses and gets stuck in a loop of formatting and syntactic errors when evaluated on SWE-rebench. The behavior reflects poor instruction following abilities of the model with limited capacity to correct its course based on feedback. This failure mode is common among low-performing models in our benchmark, but it is particularly pronounced in the Qwen2.5-Coder, where such behavior frequently prevents meaningful progress toward a solution.

It seems there is still an issue with multiple commands being extracted. Let's ensure that we only include a single command block to modify the get_adapted_type method in generators.py.

Here is the single command block again:

```
edit 183:183 << 'EOF'
    def get_adapted_type(self, coltype: TypeEngine) ->
        TypeEngine:
        if isinstance(coltype, TypeDecorator) and not
            hasattr(coltype.__class__, 'impl'):
            log.warning(f'TypeDecorator implementation {
                coltype.__class__.__name__} lacks a
                class-level variable \'impl\'. Setting
                to default type TEXT.'
            coltype.__class__.impl = TEXT # Set to
                default type TEXT at the class level
        try:
            supercls = self.get_super_class(coltype)
            if supercls is not None:
                new_coltype = coltype.adapt(supercls)
                if new_coltype.__class__ is not coltype.
                    __class__:
                    return new\_coltype
        except AssertionError as e:
            log.error(f'Error adapting type {coltype.
                __class__.__name__}: {e}. Using original
                 type.')
            return coltype
        return coltype
EOF
```

```
</agent> <environment>
File updated. Please review the changes and make sure
they are correct (correct indentation, no duplicate
lines, etc). Edit the file again if necessary.

[File: /sqlacodegen/sqlacodegen/generators.py (1016
    lines total)]
...
(Current directory: /sqlacodegen, current file: /
    sqlacodegen/sqlacodegen/generators.py)
bash-$
```

## K  Dataset schema and fields

We describe task instances using SWE-bench schema with a few extra annotations related to license type, dependency versioning and task quality. Key distinctions and additions compared to the standard SWE-bench schema include

- **License information** (license_name)**:** A new field specifying the open-source license of the source repository.

- **Enhanced metadata (`meta`):** This JSON dictionary now includes richer information beyond "lite" criteria compliance, such as the specific commit type used for patch generation (`commit_name`), the number of modified files (`num_modified_files`), and LLM-predicted scores (`llm_score`) for task difficulty, issue clarity, and test quality.
- **Pinned dependencies (`requirements, environment`):** Two new fields storing exact dependency versions (e.g., from `pip freeze` or `conda env export`) captured after successful environment setup, ensuring high reproducibility.

Table 5 lists all metadata fields included with each SWE-rebench task instance.

## L  Data collection funnel and potential enhancements

Our automated pipeline processes a large volume of raw data through several stages to curate the final SWE-rebench dataset. Table 6 summarizes the data flow, showing the approximate input and output sizes at each key stage, along with acceptance rates. Understanding this funnel helps identify bottlenecks and areas for future improvement to increase the yield of task instances.

Several strategies could be employed to potentially increase the yield at various stages of this pipeline:

- **Expanding language scope:** Currently, the pipeline focuses exclusively on Python projects. Extending support to other popular programming languages (e.g., JavaScript, Java, C++) would significantly broaden the pool of potential repositories and tasks.
- **Generalizing task sources**
  - **Beyond issue-linked PRs:** Instead of strictly requiring a PR to be linked to a formal issue, we could consider general pull requests that describe a problem and solution directly within the PR description. This would capture a wider range of code changes.
  - **Using tasks without pre-existing tests:** For PRs that solve a problem but do not include new or modified tests, an LLM could potentially be employed to generate test cases based on the issue description and the provided solution patch. This would require validation of LLM-generated tests.
- **Improving dependency installation:** The automated dependecy installation stage, with a current success rate of approximately 14%, represents a bottleneck. Substantial gains could be achieved by further improving the LLM for generating installation recipes (e.g., through more extensive fine-tuning, enhanced prompting, or employing more capable base models). Additionally, for repositories that yield a high number of task instances or are particularly popular, a more resource-intensive interactive agent could be selectively deployed to handle complex installation scenarios.

## M  SWE-rebench dataset statistics

Table 7 presents key statistics of the SWE-rebench dataset, offering insights into characteristics of its tasks. These statistics, computed across all 21,336 instances, include measures of task description length, code modification size, test suite changes, and LLM-assessed quality scores.

On average, task instances in SWE-rebench feature issue descriptions of approximately 142 words, with 75% of tasks having descriptions of 173 words or fewer. This suggests that most tasks have concise problem statements. In terms of code changes, tasks typically involve modifications to around 3-4 files (mean `Files Edited` is 3.46), with a 75th percentile of 4 files. The number of added lines averages around 97, while the 75th percentile is 76 lines, indicating that the majority of solutions are not excessively large.

Regarding test dynamics, the average number of tests transitioning from fail to pass (F2P Tests) is approximately 15. The 75th percentile for F2P tests is 5, suggesting that many tasks have a focused set of critical tests verifying the fix. The average number of tests that consistently pass before and after the patch (P2P Tests) is about 86, contributing to an average total of around 105 tests involved per task.

The LLM-assessed quality scores (where lower scores generally indicate higher quality or lower difficulty) show that, on average, tasks are perceived as having relatively low difficulty (mean

Table 5: SWE-rebench dataset schema.

| Field name | Type | Description |
|---|---|---|
| `instance_id` | str | A formatted instance identifier, typically `repo_owner__repo_name-PR_number`. |
| `patch` | str | The gold solution patch (code changes from the PR, excluding test files) that resolved the issue. |
| `repo` | str | The repository `owner/name` identifier from GitHub. |
| `base_commit` | str | The commit hash representing the repository's HEAD before the solution PR was applied. |
| `hints_text` | str | Comments made on the issue before the creation of the solution PR's first commit. |
| `created_at` | str | The creation timestamp of the pull request (ISO format). |
| `test_patch` | str | A patch containing only changes to test files contributed by the solution PR. |
| `problem_statement` | str | The concatenated title and body of the GitHub issue. |
| `version` | str | The normalized project version (e.g., "1.2") used for grouping and environment setup. |
| `environment_setup_commit` | str | The specific commit hash used as a basis for environment setup and dependency installation for this task's version group. |
| `FAIL_TO_PASS` | list[str] | JSON list of test identifiers that failed before and passed after applying the solution patch. |
| `PASS_TO_PASS` | list[str] | JSON list of test identifiers that passed both before and after applying the solution patch. |
| `meta` | dict (JSON) | A dictionary containing extended metadata. Includes: `commit_name`: (str) 'head_commit' or 'merge_commit' used for patch generation. `failed_lite_validators`: (list[str]) List of reasons an instance is not "lite". `has_test_patch`: (bool) Whether a test patch exists. `is_lite`: (bool) Whether the instance meets "lite" criteria. `num_modified_files`: (int) Number of files changed by the solution patch. `llm_score`: (dict) LLM-predicted scores: `difficulty_score`: (int) Predicted task difficulty. `issue_text_score`: (int) Predicted issue clarity. `test_score`: (int) Predicted test patch correctness. |
| `license_name` | str | The SPDX identifier or common name of the repository's license (e.g., "MIT", "Apache-2.0"). |
| `install_config` | dict (JSON) | A dictionary with the configuration used for automated environment setup. Includes fields like `python` version, `install` command, `test_cmd`, dependency file paths, etc. |
| `requirements` | str | A string containing the frozen Python dependencies (e.g., output of `pip freeze`) for the specific environment. |
| `environment` | str | A string containing the full environment specification (e.g., output of `conda env export`) for the specific environment. |

Table 6: Summary of the data collection funnel for SWE-rebench.

| Stage | Input Size | Output Size | Acceptance Rate |
|---|---|---|---|
| Match PRs & Issues | ∼10M PRs, 6M issues, 32K repos | ∼450K task candidates | ∼5% (of PRs) |
| Filter Tasks | ∼450K candidates | ∼150K filtered tasks | ∼33% |
| Install & Validate | ∼150K tasks | ∼21K valid tasks | ∼14% |
| Annotate | ∼21K tasks | ∼21K annotated tasks | ∼100% |

Table 7: SWE-rebench dataset statistics.

| Metric | Mean | p50 | p75 | p95 |
|---|---|---|---|---|
| Issue Length (words) | 141.67 | 91 | 173 | 412.25 |
| Files Edited | 3.46 | 2 | 4 | 10 |
| Lines Edited | 142.17 | 37 | 112 | 500 |
| Fail-to-Pass Tests | 14.56 | 2 | 5 | 37 |
| Pass-to-Pass Tests | 85.81 | 22 | 64 | 351 |
| Total Tests | 105.43 | 31 | 82.25 | 428 |
| Difficulty Score | 1.13 | 1 | 2 | 2 |
| Issue Text Score | 1.04 | 1 | 2 | 3 |
| Test Score | 1.38 | 2 | 2 | 3 |

`Difficulty Score` is 1.13) and clear issue descriptions (mean `Issue Score` is 1.04). The 75th percentile for these scores is 2, reinforcing that a majority of tasks are considered well-defined and not overly complex by the automated assessment model. These statistics illustrate the typical characteristics of tasks in SWE-rebench, highlighting their suitability for training and evaluating software engineering agents.

