# OpenReview forum: "SWE-rebench: An Automated Pipeline for Task Collection and Decontaminated Evaluation of Software Engineering Agents"
_NeurIPS.cc/2025/Datasets_and_Benchmarks_Track — NeurIPS 2025 Datasets and Benchmarks Track poster_

### Official Review · Reviewer_kvG9 · 2025-06-23

**Rating:** 5
**Confidence:** 2

**Summary:**

This paper proposes a new benchmark, named SWE-rebench, for the evaluation of software engineering (SWE) LLM agents. The primary contributions are: 1) a pipeline that enables scalable data collection; and 2) a public dataset of over 21,000 interactive Python tasks and a continuously updated, decontaminated benchmark. The authors show evidence that performance on their dynamic benchmark decreases substantially compared to existing static benchmarks, indicating that previously reported model scores may be inflated. This underscores the importance and value of their proposed benchmark in providing a more realistic and robust benchmark.

**Dataset Code Accessibility:**

Yes

**Dataset Code Comments:**

The authors have made the code and data for the proposed benchmark publicly available on GitHub and HuggingFace.

**Ethical Comments:**

Overall, the authors have responsibly addressed the ethical aspects of their work and been transparent about its potential societal impacts.

**Ethical Considerations:**

No, there are no or only very minor ethics concerns

**Final Justification:**

After reviewing the authors’ rebuttal, I remain positive in my evaluation and recommend acceptance. For the final version, I encourage the authors to incorporate the clarifications provided in the rebuttal. I have no further concerns.

**Limitations Weaknesses:**

1. Since the pipeline relies on an LLM to perform key tasks, like setting up the environment and performing quality checks, reproducing the whole pipeline can be difficult. This is because the outputs of the LLM may vary from run to run.

2. Although the authors have evaluated several well-known LLMs, it would greatly strengthen the work to include models like GPT‑4o, Gemini 2.5‑Flash, and Claude 4.

3. It would be helpful if the authors provided a more detailed discussion of the failure cases, specifically, why certain tasks fail during validation. Exploring these would give readers a clearer and deeper understanding of the paper’s results.

**Strengths Contributions:**

1. The automated, continuously updated pipeline is a major contribution to the field. Not only is it technically novel, but it also addresses an urgent need in LLM research.

2. Tables 1 and 2 provide compelling evidence that data contamination inflates LLM performance, underscoring the necessity of maintaining a continuously updated benchmark.

3. The benchmark has been publicly released with comprehensive, clear documentation, making it easily accessible for anyone working in the field.

---

> ### Author Rebuttal · Authors · 2025-07-30
>
> We thank the reviewer for the thoughtful and detailed feedback. We address the points below:
>
> We acknowledge the concern about reproducibility due to reliance on LLMs for key steps such as environment setup and quality checks. To mitigate variability, we incorporate safeguards such as allowing the LLM up to three attempts for dependency installation and selecting the first successful configuration. This substantially improves stability and reproducibility across runs.
>
> Regarding the inclusion of additional frontier models, we fully agree that broader model coverage strengthens the benchmark. Our public leaderboard is already being actively updated and includes GPT-4.1, Claude 3.5/4 Sonnet, o3, and Gemini 2.0/2.5 Flash. We will highlight this explicitly in the final version of the paper and continue our commitment to regularly evaluating both open and proprietary models.
>
> We also appreciate the request for a deeper analysis of failure cases. In our validation process, we observed several recurring failure modes:
> - Flaky tests, where most failures were attributed to tests involving external API calls to intermittently unavailable endpoints.
> - Tests asserting exact equality of floating-point numbers, where minor differences caused by numerical precision or rounding errors led to failures.
> - Environment setup failures that frequently occurred even when the README provided clear instructions accurately parsed by the LLM; these cases were typically linked to older repositories with unpinned dependency versions or development package versions no longer available publicly.
> We will add these details of failure modes explicitly to the final manuscript to help readers better understand the limitations and potential improvements of the pipeline.

---

> > ### Comment · Reviewer_kvG9 · 2025-08-05
> >
> > Thank you for addressing my questions. After reviewing the authors’ rebuttal, I remain positive in my evaluation and recommend acceptance. For the final version, I encourage the authors to incorporate the clarifications provided in the rebuttal. I have no further concerns.

---

### Official Review · Reviewer_b5RQ · 2025-07-02

**Rating:** 4
**Confidence:** 4

**Summary:**

This paper proposes SWE-rebench, a fully automated, scalable pipeline for extracting real-world, interactive software engineering (SWE) tasks from GitHub repositories. Using this system, the authors compile over 21,000 executable, Python-based SWE tasks aimed at enabling large-scale reinforcement learning (RL) and agentic LLM development. Additionally, they introduce a continuously updated, contamination-aware benchmark—also named SWE-rebench—designed to assess LLM-based SWE agents with minimal risk of test set leakage, addressing key limitations in prior benchmarks such as SWE-bench. The benchmark incorporates standardized agent scaffolding, centralized evaluation, and provides leaderboards with decontamination transparency.

**Additional Feedback:**

1. Can the authors quantify the error rates of the LLM-predicted task quality labels on an independent.
2. What concrete steps are planned to extend SWE-rebench beyond Python, and have preliminary trials been conducted for other languages like Java or C++?

**Dataset Code Accessibility:**

Yes

**Ethical Considerations:**

No, there are no or only very minor ethics concerns

**Limitations Weaknesses:**

1. The dataset exclusively targets Python-based projects. Although the authors claim extensibility to other languages, no experimental validation beyond Python is provided. This restricts generalizability to broader SWE domains.

2. The reliance on LLM-predicted labels (e.g., for task clarity, test correctness) introduces concerns around label noise. The validation set for quality prediction is relatively small (e.g., 413 examples), which questions the robustness of downstream task filtering.

**Strengths Contributions:**

1. The paper addresses a clear and timely gap in SWE agent development: the lack of scalable, interactive, executable tasks derived from realistic software projects.

2. The automated extraction pipeline significantly reduces manual curation bottlenecks, a known limitation in existing benchmarks like SWE-Gym and SWE-PolyBench.

3. The inclusion of continuous contamination tracking is well-motivated, considering the rapid pace of LLM development and risks of benchmark overfitting.

4. Open-sourcing both the dataset, evaluation framework, and the online leaderboard aligns with reproducibility standards expected in the community.

---

> ### Author Rebuttal · Authors · 2025-07-30
>
> We thank the reviewer for the thoughtful and detailed feedback. We address the points below:
>
> We fully acknowledge the limitation that SWE-rebench currently supports only Python-based projects. However, the pipeline itself is language-agnostic and does not rely on Python-specific assumptions. To extend the benchmark to other languages, we are actively adapting the pipeline to support Go, Java, C/C++, JavaScript/TypeScript, Rust, PHP, and Ruby. The necessary changes focus primarily on three configuration parameters:
> - install: commands to build and install dependencies,
> - test_cmd: test execution command specific to the language,
> - docker_image: appropriate base Docker environment.
> We are already collecting and curating issue/PR data from non-Python repositories and applying the same filtering and quality checks. Once finalized, the expanded multi-language dataset will be released as open-source to broaden SWE-rebench’s applicability.
>
> Regarding the use of LLM-predicted labels for task clarity and test correctness, we agree that such labels may introduce noise. These labels, however, are not used to make fine-grained quality judgments but rather to flag clearly invalid or ambiguous examples. As clarified in the paper, they serve as auxiliary signals to help users filter tasks according to their own needs. In our own benchmark setup, only conservative thresholds are applied to discard unfit instances.
>
> On the question of label accuracy, we agree that quantifying the reliability of LLM-based labels is important. While we currently validate them against a subset of SWE-bench (413 examples), we acknowledge that this is limited. We plan to evaluate the labels on larger and more diverse datasets in future work to better characterize their robustness.

---

> > ### Comment · Reviewer_b5RQ · 2025-08-08
> >
> > I appreciate the authors' detailed response, which has resolved my main concerns. I am pleased to maintain my positive evaluation of this work.

---

> ### Comment · Area_Chair_J5Xq · 2025-08-06
> **AC Message: Please Participate in the Discussion Phase**
>
> Dear reviewer b5RQ,
>
> Now that the discussion phase is reaching its end, please read the authors' rebuttal and participate in the discussion ASAP to see if they have resolved your concerns.
>
> Best,
>
> Your AC

---

### Official Review · Reviewer_ymja · 2025-07-03

**Rating:** 5
**Confidence:** 4

**Summary:**

This paper introduces a novel, scalable and automated pipeline to convert real-world interactive SWE tasks from GitHub issues. The paper is well-motivated in a continuation of more scalable and diverse SWE task benchmark beyond the original SWE-Bench. The paper writing is clear and covers some important details in the data pipeline, including the data collection, automated installation, execution-based verification, and automated quality assessment. The benchmark results show interesting results in the continuous update and decontamination, as well as the stochasity-related evaluation among all the evaluated models.

**Additional Feedback:**

* What is the main failure mode of automated environment setup? Can more rounds of LLM refinement help to increase the 31% successful installation among all the repositories?
* The benchmark mentions it selects the issues and PR between 1 to 15 scripts. What is the distribution of the number of scripts? Is the level positively correlated with the number of changed files for the existing langauge models?
* Following the first question, the authors mentioned in Section 2.4 that the fine-tuned Qwen 2.5-72B can serve as a label to facilitate difficulty control and is more precise than heuristics. Do you have any statistics on the SWE-Bench Verified to demonstrate the difference in this finetuned LLM-based critic and the heuristics, e.g. length of issues or the number of changed files?
* Can we finetune the LLM critic used for quality assessment with larger dataset beyond SWE-Bench Verified, e.g. the SWE-rebench itself based on the execution feedback?
* The authors claim that benchmark is decontaminated. In Table2, how can we identify the source for the drop between SWE-Bench Verified and SWE-rebench? Is it mainly from a generally harder set of tasks, or indeed some contamination on the SWE-Bench data?
* Are there any insights to select pass@5 as the main metrics? Maybe evaluate the impact of various N for pass@N will be helpful.

**Dataset Code Accessibility:**

Yes

**Ethical Considerations:**

No, there are no or only very minor ethics concerns

**Final Justification:**

The rebuttal addresses my previous concern about the LLM-based critic and the level filtering heuristics. I will keep my positive evaluation.

**Limitations Weaknesses:**

* Some claims are less supported empirically, e.g. the authors claim in line 213, "the number of changed files can be misleading" for the task difficulty. However, there is no quantitative studies on how the existing LLM performs under different number of changed files in SWE-rebench.
* Based on the report weighted F1 score in Section 2.4, especially the Test Patch Correctness, the LLM critic cannot give a clear classification on this. It is questionable whether the finetuned LLM-based critic can be a good surrogate verifier for RL finetuning, as the reward signal might be too noisy.
* Currently, the benchmark only supports Python as the coding language. It will be more interesting to see some extension into other languages to test the multi-domain coding skills of the LLM agents.

**Strengths Contributions:**

* The paper is well-motivated and clearly written, demonstrating both high-level motivation in data selection, and the detailed data pipeline with good transparency.
* The leaderboard includes both open-source and closed-source models and give a scalable and interesting comparisons. The benchmark can further serve for RL finetuning of the LLM agents.
* The design principles include an evaluation of standard error, which accounts for the stochasity of the models, making the evaluation results more comprehensive and meaningful.

---

> ### Author Rebuttal · Authors · 2025-07-30
>
> We thank the reviewer for the thoughtful and detailed feedback. We address the points below:
>
> Regarding the claim that the number of changed files can be misleading as a heuristic for task difficulty, we agree that empirical support was needed. We have since analyzed DeepSeek-v3-0324 performance on SWE-rebench leaderboard tasks (Jan–July) segmented by the number of changed files:
> - 1 file: 28.6% ± 0.8% (201 problems)
> - 2 files: 20.6% ± 0.4% (144 problems)
> - ≥3 files: 17.5% ± 2.2% (64 problems)
> These results confirm that difficulty increases with the number of files changed. We will include this analysis in the final paper.
>
> On the fine-tuned LLM critic used for test patch correctness: we clarify that its role is not to serve as a verifier for RL training, but as a lightweight filter to flag clearly invalid examples during curation. As noted in the paper: “While this automated assessment is not perfect, these labels offer users a means to filter the dataset and select task instances according to their specific criteria.”
>
> We agree that expanding to other programming languages would significantly enhance the benchmark. This is an active area of development. We are currently working on supporting additional languages including Go, Java, C/C++, JavaScript/TypeScript, Rust, PHP, and Ruby. These extensions will be released as open-source once completed.
>
> On the environment setup failure modes: the primary limitation is the lack of interactivity. Currently, the pipeline performs only a single round of LLM-based refinement. If a second execution fails (e.g., due to new dependency errors), the system cannot recover. We believe adding iterative refinement via interactive agents could significantly improve the current 31% success rate and plan to explore this in future work.
>
> Concerning the correlation between the number of scripts modified and task difficulty, we confirm a positive correlation, as shown above. The distribution of the number of changed files across the full SWE-rebench dataset is as follows: `1: 6497, 2: 4727, 3: 2944, 4: 1918, 5: 1285, 6: 878, 7: 702, 8: 553, 9: 464, 10: 348, 11: 242, 12: 250, 13: 195, 14: 170, 15: 138.`
>
> The idea of comparing the LLM critic with heuristic difficulty measures is valuable. While not included in this version, we consider this a promising direction for future benchmark analysis.
>
> As for fine-tuning a critic directly on SWE-rebench: this is theoretically possible, but would require additional annotations beyond execution feedback, such as human or LLM-as-a-judge evaluations. Our current critic is fine-tuned on SWE-bench Verified because of its high-quality human-labeled data. We address this problem in our subsequent work.
>
> Regarding the performance drop from SWE-bench Verified to SWE-rebench: we agree that increased task difficulty is a contributing factor. However, some model behaviors, e.g., similar performance of DeepSeek-1226 and 0324 on SWE-rebench, but divergent scores on SWE-bench Verified, suggest that contamination may be a factor in the latter. We will emphasize this analysis explicitly in the final version of the paper.
>
> Finally, we selected Pass@5 as a practical balance between evaluation cost and meaningful assessment of model capabilities. While evaluating larger Pass@N values (e.g., Pass@10) would provide additional insight, they also increase costs linearly. Nonetheless, we appreciate the suggestion and will consider reporting extended results where feasible.

---

> > ### Comment · Reviewer_ymja · 2025-08-05
> >
> > We thank the authors for providing this insightful feedback, which addresses my previous concern about the LLM-based critic and the level filtering heuristics. I will keep my positive evaluation.

---

### Official Review · Reviewer_N62b · 2025-07-06

**Rating:** 5
**Confidence:** 4

**Summary:**

This paper introduces swe-rebench, which automatically mines out verifiable, executable environments for software engineering training and evaluation. The size of the dataset is over 21K and require minimal human efforts to update. The benchmark offers a contamination (since they can be very recent) and standard protocol for evaluating language models on software engineering capability.

**Dataset Code Accessibility:**

Partly

**Dataset Code Comments:**

I see the code for running resulted swe-rebench dataset, but will the automated environment curation pipeline be opened as well?

**Ethical Considerations:**

No, there are no or only very minor ethics concerns

**Final Justification:**

author's clarification resolved most of my concerns and this is a concrete work that can bring value to the community

**Limitations Weaknesses:**

1. While the authors claims in the abstract that the environments are "suitable for reinforcement learning of SWE agents at scale", there's minimal empirical evidence to support this. The reviewer agree that conceptually this should be true.
2. While standardizing the agent framework sounds promising in standardizing the evaluation protocol, the optimal framework and prompts might vary from model to model, it's unclear to me how much the current evaluation protocol is evaluating model's performance compared to model's fitness to the specific agent framework.
3. In addition to open models, it would be helpful to also report performance on proprietary frontier models to give a more complete picture.

**Strengths Contributions:**

1. The live-updating benchmark can be a helpful resource to the community in better evaluating LM agents.
2. The method -- llm-assisted environment extraction and validation is important and helps the community to move towards more scalable ways to collect training/evaluation environments.

---

> ### Author Rebuttal · Authors · 2025-07-30
>
> We thank the reviewer for the thoughtful and detailed feedback. We address the points below:
>
> Regarding RL suitability, we acknowledge the reviewer’s concern regarding empirical evidence. While full-scale RL experiments are beyond this paper’s scope, SWE-rebench is explicitly designed to support them. Each task provides an interactive environment and a clear success/failure signal via test execution. Moreover, concurrent work (e.g., DeepSWE, Sky-RL-agent) already demonstrates the effectiveness of similar task structures for RL. We will cite these works in the final version.
>
> Regarding the comment about standardized evaluation and model’s fitness to agent framework, we agree that model-specific prompt or scaffolding tuning can lead to gains. To reduce such bias, we deliberately chose SWE-agent-like scaffolding that imposes minimum restrictions on the model’s behavior, together with simple prompt structure. We believe this sets a fair baseline for all the models.
>
> As for proprietary model evaluation, we share the reviewer’s view on the value of evaluating frontier models. Our public leaderboard already includes GPT-4.1, Claude 3.5/4 Sonnet, Gemini 2.0/2.5 Flash, and o3. We will highlight this in the final version of the paper and remain committed to regularly updating the leaderboard with both open and proprietary models.
>
> Regarding environment curation code, we understand the request for making our automated environment construction code publicly available. Due to heavy reliance on distributed infrastructure (TractoAI) for large-scale processing, directly open-sourcing the entire pipeline is challenging. However, we provide full transparency through detailed descriptions and prompt templates (Appendix B.1–B.4). To improve reproducibility, we will explore ways to provide a simplified demonstration setup for small-scale usage.

---

> > ### Comment · Reviewer_N62b · 2025-08-01
> >
> > Thanks for the clarifications. it would be great if you can open source an implementation, even if it's small-scale.
> > I think this is a concrete work. I am raising my score to 5.

---

### Note · Authors · 2025-08-13

We thank the reviewers for their commentaries. Here we will write what we have already done and the concrete actions we will take for the camera-ready version and subsequent releases.

1. The dataset is currently the most downloaded on Hugging Face this month (~3M downloads). We refresh the benchmark monthly; public updates attract strong community engagement (e.g., a recent post reached ~75k views on Reddit’s r/LocalLLaMA).

2. Our leaderboard includes GPT-5, Claude 3.5/4 Sonnet, and Gemini 2.5 Pro, alongside leading open models; we will add this in the final version.

3. While this paper focuses on Python, the pipeline is language-agnostic. We are actively adapting it for Go, Java/JS/TS, C/C++, Rust, PHP, and Ruby by swapping install, test_cmd, and docker_image. We will expand the dataset to other languages in future releases.

4. LLM-predicted labels are auxiliary filters used with conservative thresholds. In our benchmark setup, only conservative thresholds are applied to discard unfit instances.

5. We added a quantitative study showing performance decreases with more changed files (details in the rebuttal) and will integrate these statistics into the final version.

6. We will provide more information about the failed task instances during validation (flaky external calls, brittle floating-point equality, unpinned/obsolete dependencies) and discuss mitigation strategies.

7. We will reference all relevant papers published during that time.

We will incorporate the above analyses and artifacts in the final version and upcoming releases.

---

### Decision · Program_Chairs · 2025-09-18

**Decision:**

Accept (poster)

**Comment:**

This paper introduces SWE-rebench, a scalable pipeline for collecting real-world interactive software engineering (SWE) tasks from GitHub, yielding a dataset of over 21k tasks and a decontaminated benchmark for SWE agents. The strength of the work lies in tackling contamination issues in existing benchmarks and providing a large-scale, continuously updated dataset relevant for training and evaluating LLM-based agents. At the very end, all reviewers incline to accept this paper. Hence, I would recommend acceptance.